



# A robust data cleaning procedure for eddy covariance flux measurements

Domenico Vitale[1], Gerardo Fratini[2], Massimo Bilancia[3], Giacomo Nicolini[1], Simone Sabbatini[1], and Dario Papale[1,4]

[1]Department for Innovation in Biological, Agro-Food and Forest Systems (DIBAF), University of Tuscia, via San Camillo de Lellis, 01100 Viterbo, Italy
[2]LI-COR Biosciences Inc., Lincoln, Nebraska 68504, USA
[3]Ionian Department of Law, Economics and Environment, University of Bari Aldo Moro, Via Lago Maggiore angolo Via Ancona, 74121 Taranto, Italy
[4]Centro Euro-Mediterraneo sui Cambiamenti Climatici (CMCC), 01100 Viterbo, Italy

**Correspondence:** domvit@unitus.it, domvit@pec.it

**Abstract.** Integration of long-term eddy covariance (EC) flux datasets over regional and global scales requires high degree of comparability of flux data measured at different stations, which entails not only similar-performing instrumentation and their appropriate deployment, but also standardized and reproducible data processing and quality control (QC) procedures. This work focuses on the latter topic and, in particular, on the development of a robust data cleaning procedure. The proposed
strategy includes a set of tests aimed at detecting the presence of specific sources of systematic error in the data, as well as an outlier detection procedure aimed at identifying aberrant flux values. Results from tests and outlier detection are integrated in such a way as to leave a large degree of flexibility in the choice of tests and of test threshold values without losing in efficacy and, at the same time, to avoid the use of subjective criteria in the decision rule that specifies whether to retain or reject flux data of dubious quality. Tests development was rooted on advanced time series analysis techniques that consider the
stochastic properties of both raw, high-frequency EC data and of flux time series, such as complex dynamics, high persistence and possible presence of stochastic trends. The performance of each proposed test is evaluated by means of Monte Carlo simulations on synthetic datasets, whereas their impact on observed times series was evaluated on a selection of EC datasets distributed by the ICOS research infrastructure. Simulation results evidenced that the proposed tests have a better performance compared to alternative existing QC routines, showing lower false positive and false negative error rates. The application of the
proposed tests on real datasets led to an effective cleaning of EC flux data retaining the maximum number of good quality data. Although there is still room for improvement, in particular with the development of new QC tests, we think that the proposed data cleaning procedure can serve as a basis towards a unified QC strategy for EC datasets which $i$) includes only completely data-driven routines and is therefore suitable for automatic and centralized data processing pipelines, $ii$) guarantees results reproducibility and $iii$) is flexible and scalable to accommodate new and additional tests that makes the approach also suitable
for other greenhouse gases.



# 1 Introduction

In the last decades, the number of Eddy Covariance (EC) stations for measuring biosphere-atmosphere exchanges of energy and greenhouse gases (mainly $CO_2$ and $H_2O$, followed by $CH_4$ and $N_2O$) increased worldwide, contributing to expand regional (e.g. ICOS, AmeriFlux, NEON, TERN) and global (e.g. FLUXNET) monitoring networks. Integration of long-term flux datasets over regional and global scales enables the evaluation of climate-ecosystem feedbacks and the study of the complex interactions between terrestrial ecosystems and the atmosphere. The use of the EC technique involves a set of complex choices. Selection of the measurement site and of the instrumentation, design of the data acquisition strategy, deployment and maintenance of the EC system as well as design of the data processing pathway are only some examples of such choices. Over time, the EC community has developed guidelines and best practices aimed at "standardizing" the methodology, with the overarching goal of increasing comparability and integrability of flux datasets across different stations, thereby improving robustness and accurateness of resulting synthesis, analysis and models. Examples of efforts in this direction can be found in the EC handbooks by Lee et al. (2005) and by Aubinet et al. (2012), in publications describing standardized EC systems for entire networks (e.g. Franz et al., 2018) and in software intercomparisons aimed at reconciling the complex EC processing chain and explain/quantify any discrepancy (e.g. Mauder et al., 2006; Fratini and Mauder, 2014; Metzger et al., 2017). As part of this effort, large regional networks have recently invested significant resources in the definition of protocols and measurement methods (e.g. Sabbatini and Papale, 2017; Rebmann et al., 2018; Nicolini et al., 2018) and in the development of centralized data processing pipelines (e.g. Sabbatini et al., 2018).

Integral part of the EC method is the definition of Quality Assurance (QA) and Quality Control (QC) procedures. Quality assurance refers to the set of measures aimed at preventing errors and therefore concerns design of the experimental setup, selection of the site, choice of instrumentation and its deployment, maintenance scheduling. Quality control, instead, refers to the ensemble of procedures aimed at: (1) identifying and eliminating errors in resulting datasets (i.e. data cleaning) and (2) characterizing the uncertainty associated to flux measurements. This paper is concerned with the definition of QC procedures for EC datasets, in particular for error identification and data cleaning.

In the context of EC, a thorough QC scheme should aim at detecting errors caused by instrumental issues as well as by violations of the assumptions underlying the method (see Foken et al., 2012b; Richardson et al., 2012), which inevitably occur during the measuring campaign. Surface heterogeneities and anisotropies, occurrence of poorly developed turbulence conditions, advection fluxes as well as instrumental malfunctions (leading to e.g. spikes or discontinuities in the raw data), miscalibrations or operation below the detection limit of the instruments are common sources of systematic errors introducing biases in resulting fluxes. Systematic errors can also occur during the data processing stage, due to inappropriate choices of the processing options, poor parameterization of corrections (e.g. a poor estimation of spectral attenuations in an EC system) or to edge cases that lead algorithms to diverge (e.g. when the denominator of a ratio tends to zero). An effective QC procedure should in principle be able to identify all such occurrences and hence allow the elimination of the corresponding flux values.

It is worth noting that EC measurements, as any measurement process, are also subject to a number of sources of unavoidable random errors causing noise in flux data, due for example to sampling a 3D stochastic process at a single point, or due to



the finite precision of the measuring devices, or to the variability of the source area (the so-called flux footprint) within the flux averaging time scale. By definition, random error cannot be eliminated in single-measurement experiments such as EC measurements. However, their effect can be minimized by careful QA procedures (e.g. selection of station location and of instrumentation, relative to the intended application) and quantified by characterizing the random error distribution through an appropriate probability density function (PDF), most commonly assumed to be well approximated by a Normal or Laplace

distribution (e.g. Richardson et al., 2008; Lasslop et al., 2008; Vitale et al., 2019). The characterization of random error distribution is beyond the scope of the current paper, but we will make some considerations in this respect as relevant to our objectives.

Following Richardson et al. (2012), in order to define a modelling framework suitable for the representation of a variable $y$ observed with error (e.g. an half-hourly flux time series), we stipulate that the observed value at time $t$, $y_t^{\text{obs}}$, is related to the

true value of the variable, $y_t^{\text{true}}$, via:

$$y_t^{\text{obs}} = y_t^{\text{true}} + e_t \qquad (1)$$

where $e_t$ is an error term that can be further specified as:

$$e_t = \sum_i \beta_{i,t} + \varepsilon_t = \beta_t + \varepsilon_t \qquad (2)$$

where $\beta_t$ represents the total systematic error given by the sum of all individual systematic error components, $\beta_{i,t}$, biasing the

variable, and $\varepsilon_t$ is the random measurement error. The standard deviation characterizing the distribution of random errors, $\sigma_\varepsilon$, provides a measure of the random uncertainty.

With this formalization in mind, QA procedures are aimed at preventing all sources of systematic errors ($\beta_i \to 0 \; \forall i$) and at making sure $\sigma_\varepsilon$ is as small as possible, while QC procedures are aimed at detecting and rejecting any $y_t^{\text{obs}}$ data points, for which the effect of any systematic error on flux estimates is not negligible (i.e. $\beta_t \neq 0$).

To avoid confusion, it is worth stressing the difference between random and systematic errors affecting EC flux measurements. By definition, systematic errors in an individual flux measurement are those due to sources that are avoidable, at least in principle. As an example, avoidable instrumental errors can lead to spikes in the raw, high-frequency time-series. Such spikes, regardless of their distribution and amplitude, lead to a biased flux and are therefore to be regarded as systematic errors. On the contrary, random errors are by definition unavoidable and undetectable. It follows from those definitions that long-term biases

in flux time-series (say, a systematic underestimation) can only be caused by systematic errors. However, systematic errors do not necessarily lead to long-term biases, because they can vary with time, both in absolute value and relative to the true flux values. Random errors, instead, never lead to long-term biases (Moncrieff et al., 1996; Richardson et al., 2012).

QC procedures developed to identify EC fluxes affected by significant errors can be broadly classified in partially and completely data-driven (or automatic) approaches. The former entail at least some degree of subjective evaluation in the decision

process. They rely on the ability of the analyst to make a final call on whether a data point should be retained or rejected, allowing the researcher to exploit the accumulated knowledge about the site and the dataset, in order to discern what is physically or ecologically implausible. Such call is usually made on the basis of some preliminary error detection algorithm and,





in practice, is typically performed via visual inspection. As an example, Vickers and Mahrt (1997) proposed a suite of tests to detect problems with high-frequency raw data. There, each test results in a label for the time-series, such as retain or *potentially reject*, and the investigator is required to make the final call on the latter. The drawbacks of such procedures is that subjective evaluation unavoidably introduces individual biases, which weaken the robustness and the objectivity of results. In addition, the fact that subjective evaluation is usually performed via visual inspection strongly affects traceability of the data processing history, severely hindering reproducibility of resulting datasets. Completely data-driven procedures, on the contrary, sacrify the benefit of first-hand knowledge of the dataset to gain high levels of reproducibility and objectivity, by means of fully automated QC algorithms and decision trees. An important advantage of such procedures is that they strongly reduce the possibility of introducing selective subjective biases when cleaning datasets across multiple stations, thereby contributing to the standardization we referred to at the beginning. Moreover, fully automated algorithms are preferable when the processing involves the ingestion of massive data and the use of visual inspection becomes prohibitive because of extremely time consuming. However, the construction of a completely data-driven procedure that accounts for and exploits all available knowledge of the system is complex and requires care in testing development to prevent high false positive rates.

Usually, a QC procedure is comprised of multiple routines, each of which evaluates data with respect to a specific source of systematic error. In the case of EC, we therefore have routines to identify instrumental issues, severe violations of the method assumptions or issues with the data processing pipeline. Obviously, there may be several routines for each category, e.g. routines to look at specific types of instrumental issues (e.g. spikes, reached detection limit, implausible discontinuities in the time-series, etc.). Once a set of QC tests has been selected, results of these tests must be combined, in order to derive an actionable *label* to reject or retain individual flux data. One of the most popular QC classification schemes was proposed by Mauder and Foken (2004) and was more recently adopted, with some modifications, by Mauder et al. (2013). This classification scheme establishes a data quality flag on the basis of a combination of results from two tests, proposed by Foken and Wichura (1996), referred to as *Instationarity* and *Integral Turbulence Characteristics* tests. Results of the two tests are combined based on a predefined policy to derive the final classification, which assigns the overall flag 0 to *high-quality* data, 1 to *intermediate quality* data, and 2 to *low-quality* data. Recommendations are that: ($a$) low-quality data should be rejected; ($b$) the use of intermediate quality data should be limited to specific applications, e.g. long-term budget calculation, while it should be avoided in more fundamental process studies; and ($c$) high-quality data can be used for any purposes. Other quality classification schemes of this sort have been proposed and used (e.g. Kruijt et al., 2004; Göckede et al., 2008; Thomas et al., 2009).

Two aspects of this approach to QC classification are of concern. Firstly, the combination of the results of individual tests into an overall flag appears to be somewhat arbitrary and no directives are provided - nor they are easily imagined - as to how to extend the combination to integrate results of additional or alternative tests. More fundamentally, with the methods above the final aim of the process is to attach a *quality* statement to the data via a flag, as opposed to identifying data points affected by severe errors, therefore confounding the two processes - that we deem distinct - of cleaning the dataset and of characterizing its quality. We suggest, instead, that a *data cleaning* procedure for EC datasets should exclusively aim at identifying flux values affected by errors large enough to warrant their rejection. It should, therefore, lead to a binary classification, such as *retain* or *reject*. As for assigning a quality level to the retained data, we propose that the random uncertainty is the appropriate





metric. In fact, as a general principle, the larger the random uncertainty, the larger the amount of measurement error and, consequently, the lower the quality of the data (for an in depth discussion of the random uncertainty in EC, see e.g. Richardson
et al. (2012), and references therein). The link between data quality and random uncertainty has already been investigated by Mauder et al. (2013). Using the turbulence sampling error proposed by Finkelstein and Sims (2001) as a measure of the flux random uncertainty, they found that highest-quality data are typically associated with relative random uncertainties of less than 20%, whereas intermediate quality data are typically associated with random uncertainties between 15% and 50%. However, explicitly avoiding the binning into intermediate quality and high-quality data allows us to avoid uncertain recommendations
as to what application a given intermediate quality datum should be used for. Rather, we recommend taking into account the random uncertainty throughout the data analysis and synthesis pipeline and let the application itself (e.g. data assimilation into modelling frameworks) reveal whether the level of uncertainty of each data point in the dataset matters or not.

Bearing this in mind, the aim of this paper is thus to present a robust data cleaning procedure which $i$) includes only completely data-driven routines and is therefore suitable for automatic and centralized data processing pipelines; $ii$) guarantees
results reproducibility; $iii$) is flexible and scalable to accommodate addition or removal of routines from the proposed set; $iv$) results in a binary label such as *retain* or *reject* for each data point, decoupling the data cleaning procedure from its quality evaluation.

## 2   Materials and Methods

### 2.1   The proposed data cleaning strategy

Since quantifying bias affecting half-hourly EC fluxes is not possible in the absence of reference values, the only option is to ascertain the occurrence of specific sources of systematic error which, in turn, are assumed to introduce biases in flux estimates. Therefore, the proposed data cleaning procedure includes $i$) a set of test aimed at detecting the presence of a specific source of systematic error and $ii$) an outlier detection procedure aimed at identifying aberrant flux values. As described below, results from tests and outlier detection are integrated in such a way as to leave a large degree of flexibility in the choice of tests and
of test threshold values without losing in efficacy while striving to avoid the use of subjective criteria in the decision rule that specifies whether to ultimately retain or reject flux data of dubious quality.

For each test, two threshold values are defined, designed to minimize false positive and false negative error rates. Comparing the test statistic with such thresholds, each test returns one of 3 possible statements:

- *SevEr*: if the test provides strong evidence about the presence of a specific source of systematic error.

- *ModEr*: if the test provides only weak evidence about the presence of a specific source of systematic error.

- *NoEr*: if the test does not provide evidence about the presence of a specific source of systematic error.

Threshold values can be set either based on the sampling distribution of the test statistic (e.g. when the sampling distribution of the statistic is standard normal, tabulated $z$-critical values can be used) or making use of the laws of probability or, when not





possible, by evaluating the distribution of the test statistic on large datasets and establishing those values in correspondence of
pre-fixed significance level minimizing false positive errors. Although the definition of threshold values inevitably introduces
some level of subjectivity based on domain-specific knowledge, it does not hinder the overall traceability of the process and
results reproducibility.

The following subsections detail the set of tests used in the proposed data-cleaning procedure and summarized in Table 1.
Depending on their specification, test are applied to either individual high-frequency raw time-series, or to pairs of variables,
or to their statistics over the flux averaging interval, or even to derived variables (see Secs. 2.2 for the details). For example,
instrumental problems are typically searched for on a per-variable basis, while stationarity conditions are assessed by looking
at the covariance between two variables. The test result is then inherited by all fluxes, to which the variable or variable pair
is relevant. As an example, sensible heat flux $H$ inherits the test results assigned to (at least) vertical wind speed $w$, sonic
temperature $T_s$ and the pair $w/T_s$.

The tests proposed and detailed in the following subsections have been designed and optimized for $CO_2/H_2O$ and energy
EC fluxes over natural ecosystems based on current knowledge. However in different conditions (gas species, ecosystem,
instrumentation etc.), new tests and further optimization of the ones proposed here are possibly needed. As mentioned above,
the data cleaning procedure is designed to naturally accommodate any number of tests that generate a statistic that can be
compared against threshold values.

Test results are used as inputs to the data cleaning procedure, which includes two stages (see Fig. 1). In the first stage, fluxes
that inherited at least one SevEr statement are rejected, while fluxes that inherited no SevEr statements and any number of
ModEr statements are retained. Importantly, the statements resulting from different tests (or from the same test as applied to
different variables) are not combined in any way. The rationale is that a single SevEr statement is sufficient to establish the
rejectability of a data point. Conversely, no matter how many ModEr statements are inherited by a flux data, those statements
do not accrue to the status of SevEr and hence they do not lead per se to the rejection of the measurement. This is because of
two reasons: $i.$ test independence cannot in general be guaranteed, i.e. the same source of error can affect the statistics of more
than one test; and $ii.$ nothing can be told about how different systematic errors combine, e.g. whether they tend to cumulate
or cancel out. This policy has the convenient side effect of making the application and interpretation of tests completely
independent from each other, which makes the overall procedure extremely flexible and scalable to accommodate new tests
beyond what we propose later in this paper (e.g. tests for less investigated gas species): all that is required is for the new test to
return statements such as SevEr, ModEr (if applicable) and NoEr.

In the second stage, flux data that inherited no SevEr statement are subject to an outlier detection procedure and only flux
data that are both detected as outlier and inherited at least a ModEr statement are conclusively rejected. This implies that data
points that inherited any number of ModEr statements but were not detected to be outliers, as well as outliers which showed
no evidence of systematic errors, are retained in the dataset and can be used for any analysis or modeling purposes. In other
words, only data points that provide strong evidence of error (either because of a SevEr, of because of a ModEr and being
identified as an outlier) are rejected, while peculiar data points, that would look suspicious to the visual inspection and are
possibly identified as outliers, but only inherited NoEr statements, are retained.



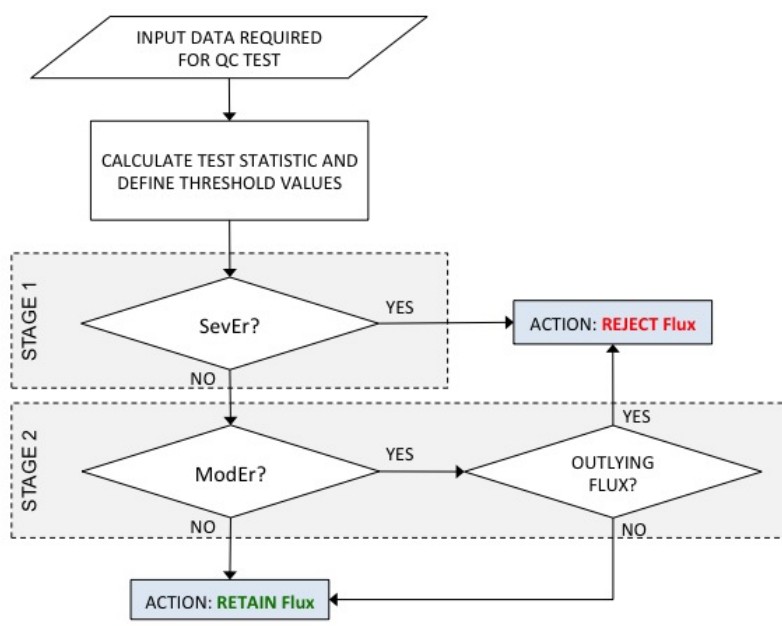

**Figure 1.** Schematic summary of the proposed data cleaning procedure. SevEr and ModEr indicating strong and moderate evidence about the presence of a specific source of systematic error, respectively.

## 2.2 Detection of sources of systematic error affecting EC datasets

The possible sources of systematic errors are divided and analysed into three categories: 1) instrumental issues, 2) poorly developed turbulence regimes and 3) non-stationary conditions.

### 2.2.1 Detection of instrumental issues

Modern EC instruments can detect and report malfunctions occurring during the measurement process via diagnostic variables. However, there are situations where a measurement is affected by an error but it is still valid from the instrument's perspective,

and for this reason it is not flagged by the instrument diagnostics. As an example, a physical quantity may have variations that are too small to detect, given the settings or the specifications of the instrument. This is the case of a time-series of temperature that varies very little during a calm, stable night; the measurements may be affected by a low-resolution problem, where the quantization of the measurement due to the intrinsic resolution of the instrument becomes evident and leads to a reduced variability of the underlying signal. In this case from the measurement perspective there is nothing wrong with the measured

quantity and diagnostics would not indicate any issues. In addition, with some instrumentation, especially older one, or often





**Table 1.** Sources of systematic error, test statistics and adopted threshold values to define NoEr, ModEr and SevEr statements. Details on how the threshold values have been set are provided in Section 2.2.

| Source of Error | Test statistic | Threshold values | | |
|---|---|---|---|---|
| | | **NoEr** | **ModEr** | **SevEr** |
| 1) EC system malfunction and disturbances | 1a) FMR - fraction of missing records (%) | $\leq 5$ | $\leq 15$ | $> 15$ |
| | 1b) LGD - longest gap duration (sec) | $\leq 90$ | $\leq 180$ | $> 180$ |
| 2) Low signal resolution | 2) $R^2$ for linear regression of CCFs estimated with original data and after removal of repeated contiguous values | $> 0.995$ | $\leq 0.995$ | $< 0.99$ |
| 3) Aberrant structural changes (e.g. sudden shift in mean, changes in variance) | Homogeneity test of fluctuations: | | | |
| | 3a) HF$_5$ - % of fluctuations beyond $\pm 5\sigma$ | $\leq 2$ | $\leq 4$ | $> 4$ |
| | 3b) HF$_1$ % of fluctuations beyond $\pm 10\sigma$ | $\leq 0.5$ | $\leq 1$ | $> 1$ |
| | Homogeneity test of differenced data: | | | |
| | 3c) HD$_5$ - % of differenced data beyond $\pm 5\sigma$ | $\leq 2$ | $\leq 4$ | $> 4$ |
| | 3d) HD$_1$ - % of differenced data beyond $\pm 10\sigma$ | $\leq 0.5$ | $\leq 1$ | $> 1$ |
| | 3e) KID - kurtosis index of differenced data | $\leq 30$ | $\leq 50$ | $> 50$ |
| 4) Poorly developed turbulence regimes | 4) Integral Turbulence Characteristics (%) by Foken and Wichura (1996) | $\leq 30$ | $\leq 100$ | $> 100$ |
| 5) Non-stationary conditions | 5) Non-stationary ratio by Mahrt (1998) | $\leq 2$ | $\leq 3$ | $> 3$ |

when collecting data in analog format, diagnostic information is simply not available (Fratini et al., 2018). It is therefore useful to devise tests to detect instrument-related situations that are likely to generate systematic errors in resulting fluxes. In the following, we describe a series of routines to identify the most frequent errors caused by instrumental problems detectable on the raw, high-frequency time-series (most of which were already discussed in Vickers and Mahrt, 1997, hereafter VM97).

**Detection of unreliable records caused by system malfunction and disturbances**

EC fluxes are calculated starting from the covariance between the vertical wind speed, $w$, and the scalar of interest computed over a specified averaging interval, typically 30 or 60 minutes. For the calculated covariance to be representative of the entire interval, the actual number of available raw data records should be close enough to the expected number (e.g. 18000 records for fluxes over 30 minutes from raw data sampled at 10 Hz). If the number of available records is too small, the corresponding

flux estimate may be significantly biased.

Data records can be unavailable for covariance computation for a variety of reasons. First, records may simply be missing because of problems during data acquisition. In addition, specific values may be eliminated if:





✓ Instrumental diagnostics flag a problem with the measurement system.

✓ Individual high-frequency data points are outside their physically plausible range or are identified as spikes (in this work
we used the despiking algorithm proposed by VM97).

✓ Data were recorded during periods when the wind was blowing from directions known to significantly affect the turbulent
     flow reaching the sonic anemometer sampling volume, e.g. due to the interference of the anemometer structure itself (C-
     clamp models) or to the presence of other obstacles.

✓ The angle-of-attack of individual wind measurements is beyond the calibration range specified by the sonic anemometer
manufacturer.

Note that the criteria above may apply to single variables, groups of variables (e.g. anemometric variables) or to entire records. Although covariances can be computed also on time series with gaps, some of the procedures involved in the typical data processing do require continuous time series (e.g. the Fast Fourier Transform to compute power spectra and cospectra, or the stationarity tests that compute statistics on sub-periods as short as 50 seconds, Mahrt, 1998). The performance of such
procedures may depend on the technique used to impute missing data (i.e. fill the gaps). It is therefore useful to establish criteria for the appropriate use of imputation procedures.

     Typically, gaps in raw time series are filled using linear interpolation. While this algorithm provides obvious computational and implementation advantages, it should be noted that it only performs satisfactorily when the time series is dominated by low-frequency components, while it can introduce biases in time series characterized by significant high-frequency components, as
it is the case with EC data. Its application should thus be limited to very short gaps. The pattern of missing data plays a role too: when gaps occur simultaneously across all variables, linear interpolation can lead to biases in resulting covariances even for short gaps. Such biases are linearly proportional to the amount of missing data, and relatively larger for smaller fluxes. These considerations apply also to other interpolation methods such as splines and Last Observation Carried Forward.

     With this in mind, indicating with FMR and LGD the fraction of missing records and the longest gap duration in time series
involved in the covariance estimation, respectively, we suggest the following classification criteria:

○ SevEr: *if* FMR > 15% *or* LGD > 180 sec.

○ ModEr: *if* 5% < FMR ≤ 15% *or* 90 sec < LGD ≤ 180 sec.

○ NoEr: *if* FMR ≤ 5% *and* LGD ≤ 90 sec.

**Detection of low signal resolution problems**

High frequency EC data can be affected by low signal resolution problems (see VM97 for a detailed description). Resolution problems are mainly caused by a limited digitalization of the signal during data acquisition, when signal fluctuations approach the resolution of the instruments. This kind of problem causes a step ladder in the distribution of sampled data and time series are characterized by the presence of repeated contiguous values. Instrumental faults can lead to a time series that remains





fixed at a constant value for a period of time (dropout), analogously introducing artificial repeated values, though with a very

different pattern of repetition. Repeated values are always to be considered an artefact since even in the (unlikely) event of a signal maintaining a constant value for an extended period of time, its measured values would still vary on account of the unavoidable random measurement error. In this particular scenario, contiguous repeated values wouldn't actually lead to a bias in the flux estimate, because neglecting the presence of random error does not affect covariance estimates. Instead when repeated values do not reflect the true dynamic of the underlying signal, they can lead to a significant bias in flux estimates.

To disentangle these two situations, we evaluate the discrepancy between the $w$-scalar cross-correlation functions (CCF) of the original time series and of the time series after removal of repeated values. If the discrepancy is negligible then the effect of repeated values on the covariance estimate is considered irrelevant. This could occur in case of small number of repeated data or if their values are still representative of the underlying signal dynamic. Conversely, when a significant discrepancy is found the covariance estimate is considered biased. To evaluate the significance of such discrepancy, we propose to use the coefficient

of determination ($R^2$) of the linear regression through the origin of the CCFs estimated at short lags ($\pm 25$ steps). The criteria used to assign a statement are as follows:

- SevEr: *if* $R^2$ <0.990.

- ModEr: *if* $0.990 \leq R^2 \leq 0.995$.

- NoEr: *if* $R^2$ >0.995.

**Detection of aberrant structural changes in time series dynamics**

EC time series can be subject to regime changes such as sudden shifts in the mean and/or in the variance. In some circumstances, those are imputable to natural causes as in cases of intermittent turbulence (Sandborn, 1959) and wind pattern change over a heterogeneous or anisotropic footprint. In other cases, such changes are artefacts generated by instrumental malfunctions. Aberrant structural changes in mean and variance due to either environmental causes or measurement artefacts have

similar effects on time series dynamics (which makes them difficult to disentangle) and lead to violation of the assumption of stationarity underlying the EC method. They should therefore be treated as sources of systematic errors. Here we propose three new tests to identify such situations, whose rejection rules is based on objective criteria but which, notably, do not discern natural changes from measurements artefacts.

The first test takes into consideration the homogeneity of the distribution of fluctuations ($y'_t = y_t - \bar{y}$, where $\bar{y}$ is the mean

level estimated according to the averaging rule adopted in the covariance calculation, e.g. block average or linear detrending) and consists in evaluating the percentage of data in the tails with respect to the bounds imposed by Chebyshev's inequality theorem. Irrespective of the PDF generating the data, Chebyshev's inequality provides an upper bound to the probability that absolute deviations of a random variable from its mean will exceed a given threshold value. As an example, it imposes that a minimum of 75% of values must lie within two standard deviations of the mean, and 89% within three standard deviations.

When the inequality is violated, it is a symptom of data inhomogeneity due to structural changes in time series dynamics (e.g. abrupt change in the mean level, sudden upward or downward trend which may introduce significantly bias in the estimation of





the mean values and, consequently, of the covariances). In the following we indicate the test statistic related to the homogeneity of the distribution of fluctuations as $\mathrm{HF}_B$, where the subscript $B$ indicates the sigma-rule adopted to define the boundary region (e.g. $\pm 5\sigma$, see later).

The second test makes use of the same rule, but evaluates the homogeneity of the distribution of first-order differenced data, $\Delta y_t = y_t - y_{t-1}$ (the corresponding statistic is denoted as $\mathrm{HD}_B$). Besides highlighting other useful properties, differencing a variable eliminates any trends present in it, whether deterministic (e.g. linear) or stochastic (Box et al., 2015) and the resulting time series is always stationary. Differencing acts as a signal filtering procedure and the transformed data can better highlight the characteristics of the measurement error process whose variance, under second-order stationary conditions, should be

constant over time. Therefore, when the inequality is violated, it is a symptom of data inhomogeneity mainly due to changes in variance (heteroskedasticity). Based on the upper bounds imposed by Chebyshev's inequality for $5\sigma$ and $10\sigma$, the first two tests are summarized in the following rules:

○ SevEr: *if* $\mathrm{HF}_5 > 4\%$ *or* $\mathrm{HD}_5 > 4\%$ *or* $\mathrm{HF}_{10} > 1\%$ *or* $\mathrm{HD}_{10} > 1\%$ (i.e., more than $4\%$ of fluctuations or differenced values are beyond the $\pm 5\sigma$ limits or more than $1\%$ of them are beyond the $\pm 10\sigma$ limits).

○ ModEr: *if* $2\% < \mathrm{HF}_5 \leq 4\%$ *or* $2\% < \mathrm{HD}_5 \leq 4\%$ *or* $0.5\% < \mathrm{HF}_{10} \leq 1\%$ *or* $0.5\% < \mathrm{HD}_{10} \leq 1\%$ (i.e., more than $2\%$ of fluctuations or differenced values are beyond the $\pm 5\sigma$ limits or more than $0.5\%$ of them are beyond the $\pm 10\sigma$ limits).

○ NoEr: *if* $\mathrm{HF}_5 \leq 2\%$ *and* $\mathrm{HD}_5 \leq 2\%$ *and* $\mathrm{HF}_{10} \leq 0.5\%$ *and* $\mathrm{HD}_{10} \leq 0.5\%$ (i.e., less than $2\%$ of both fluctuations and differences are beyond the $\pm 5\sigma$ limits and less than $0.5\%$ of them are beyond the $\pm 10\sigma$ limits).

As a robust estimate of $\sigma$, we used the Rousseeuw and Croux (1993) $Q_n$ estimator corresponding approximately to the first

quartile of the sorted pairwise absolute differences. Compared to other robust scale estimators - such as the median absolute deviation about the median (MAD) and the interquartile distance (IQD) - it is a location-free estimator, i.e. it does not implicitly rely on a symmetric noise distribution. Similar to MAD, its breakdown point is $50\%$, i.e. it becomes biased when $50\%$ or more of the data are large outliers, however, its efficiency is larger, especially when identical measurement occur, e.g. due to low signal resolution problems.

The third test consists in evaluating the kurtosis index on the differenced data $\Delta y_t$ (hereafter denoted KID test). The kurtosis index is defined by the standardized fourth moment of the variable about the mean. Because variations about the mean are raised to the power of 4, the kurtosis index is sensitive to tail values of the distribution and can therefore be used to characterize them (Westfall, 2014). Since the tails of a distribution represent events outside the "normal" range, a higher kurtosis means that more of the variance is contributed by infrequent extreme observations (i.e. anomalies) as opposed to frequent, modestly sized

deviations. The sensitivity to tail values and the application to differentiated values make the KID a very useful tool to detect a range of instrumental problems, which include abnormal changes in mean, variance, but also the presence of undetected residual spikes and drop-out events (in these situations differenced time series will show a spike in correspondence of each change-point).

To eliminate the sensitivity of the KID to the presence of repeated values (which become zeros in the differenced variable),

such values were not included in KID estimation. Bearing in mind that the kurtosis index for a Gaussian and a Laplace PDF





is equal to 3 and 6, respectively, we choose reasonably large threshold values to make sure we select only time series charac-
terized by heavy-tailed distribution as is the case of data contaminated by extreme events representative of the aforementioned
problems. Namely, we suggest that the following criteria be applied:

- SevEr: *if* KID $> 50$.

- ModEr: *if* $30 <$ KID $\leq 50$.

- NoEr: *if* KID $\leq 30$.

### 2.2.2 Detection of poorly developed turbulence regimes

One of the assumptions underlying the EC method is the occurrence of well-developed turbulence conditions. A widely used
test to assess the quality of the turbulence regime is the Integral Turbulence Characteristics (ITC) test proposed by Foken
and Wichura (1996, hereafter FW96). Based on the flux-variance similarity theory, the test consists in quantifying the relative
change of the ratio between the standard deviation of a turbulent parameter and its corresponding turbulent flux (the "integral
characteristic") with respect to the same ratio estimated by a suitable parameterization. The ITC can be calculated for each
wind component and any scalar (temperature and gas concentration), but for the purpose of data cleaning, the ITC for vertical
wind speed is used (see Foken et al., 2012b). This is defined as:

$$\text{ITC} = \left| \frac{\sigma_w/u^*}{f(\sigma_w/u^*)} - 1 \right| \cdot 100 \tag{3}$$

where $\sigma_w$ is the standard deviation of the vertical wind speed $w$, $u^*$ the friction velocity and $f(\sigma_w/u*) = c_1 \cdot ((z-d)/L)^{c_2}$ is
the parameterization as a function of the dynamic measurement height $(z-d)$ and the Obukhov length $L$, with $c_1$ and $c_2$ being
parameters varying with atmospheric stability conditions, as tabulated in Foken et al. (2012b, Tables 4.2 and 4.3).

The criteria adopted to assign SevEr, ModEr and NoEr statements are based on threshold values proposed by Mauder and
Foken (2004, Table 16). In particular:

- SevEr: *if* ITC $> 100$.

- ModEr: *if* $30 <$ ITC $\leq 100$.

- NoEr: *if* ITC $\leq 30$.

### 2.2.3 Detection of non-stationary conditions

The working equation of turbulent fluxes as the (appropriately scaled) w-scalar covariance is based on a simplification of the
mass conservation equation (see e.g. Foken et al., 2012a). One of the assumptions behind such simplification is that of stationary
conditions, which are however not always fulfilled during the measurement period. Diurnal trends, changes in meteorological
conditions (e.g. passage of clouds), boundary layer transitions, large scale variability and changes in footprint areas are only
some example of sources generating nonstationary conditions. In presence of non-stationarity, flux estimates are biased. The





magnitude and sign of the systematic error depend on the nature of the non-stationarity, on the proportion of total variability explained by trend components, and on the way (deterministic or stochastic) independent trend components interact with each other.

To avoid biases, a possible approach is to preliminarily remove the source of non-stationarity before calculating covariances. This way, the amount of data rejected for non-stationary conditions can be limited. To this end, procedures based on linear

detrending or running mean filtering are often used during the raw data processing stage. However, their application can be ineffective, for example when linear detrending is used on nonlinear trends (see the Supplementary Material (SM), Appendix B) and even expose to the risk of introducing further biases when data are truly stationary or when nonstationarity is of more complex nature (see Rannik and Vesala, 1999; Donateo et al., 2017, for a comparison of detrending methods). For this reason, an alternative approach is to remove periods identified as non-stationary.

A few tests have been proposed for EC data, of which we discuss two popular ones. A widely used statistics is based on a test introduced by FW96, based on the comparison between the covariance computed over the flux averaging period $T$ (30 or 60 minutes) and the average of covariances computed on shorter intervals $I$ (in the original proposal, 6 periods of 5 minutes each if $T$ is 30 minutes). The test statistic is defined as:

$$S_{\text{FW96}} = \left| \frac{\langle x'y' \rangle - \overline{x'y'}}{\overline{x'y'}} \right| \tag{4}$$

where $\langle x'y' \rangle = \frac{1}{I} \sum_{i=1}^{I} [\overline{x'y'}]_i$ is the mean covariance obtained by averaging covariances $[\overline{x'y'}]_i = 1, 2, ...., I$ computed on $I$ intervals, and $\overline{x'y'}$ is the covariance computed over the period $T$. According to the authors' suggestion, stationary conditions are met when $S_{\text{FW96}} \leq 30\%$.

A major problem with Eq. 4 is that when the covariance at the denominator is close to zero, the test statistic will approach infinity making the ratio unstable. As a consequence the FW96 test is disproportionately sensitive to fluxes of small magnitude

(see also Pilegaard et al., 2011).

An alternative test was proposed by Mahrt (1998, hereafter M98). This test measures nonstationarity by comparing the effects of coherent behavior with the inherent random variability in the turbulent signals. As with FW96, the time series is divided into $I$ non-overlapping intervals of 5 minutes; in this case however each interval is also divided into $J = 6$ non-overlapping sub-intervals of 50 seconds. The test statistic is defined as:

$$S_{\text{M98}} = \frac{\sigma_B}{\sigma_W / \sqrt{J}} \tag{5}$$

where $\sigma_B$ is a measure of the variability of the covariance *between* different intervals calculated as:

$$\sigma_B = \sqrt{\frac{1}{I-1} \sum_{i=1}^{I} \left( [\overline{x'y'}]_i - \overline{x'y'} \right)^2} \tag{6}$$

and $\sigma_W$ is a measure of the average variability *within* each interval $i$ and is given by:

$$\sigma_{W(i)} = \sqrt{\frac{1}{J-1} \sum_{i=1}^{I} \left( [\overline{x'y'}]_{i,j} - [\overline{x'y'}]_i \right)^2} \qquad i = 1, ..., I \tag{7}$$





with $\overline{x'y'}$ denoting the covariance computed over the flux averaging period, $[\overline{x'y'}]_i$ the covariance relative to the $i^{th}$ 5-minute interval, and $[\overline{x'y'}]_{i,j}$ the covariance relative to the $j^{th}$ 50-second subinterval within the $i^{th}$ interval. Mahrt (1998) recommendation is to reject flux data when $S_{M98} > 2$, without excluding the possibility to increase this value based on further considerations.

Respect to the FW96 test, the non-stationarity ratio by M98 is always well-behaved having the denominator strictly positive
(average of standard deviations). For this reason, the test proposed by M98 was selected for this work. Based on a performance evaluation described later in this paper (see Sect. 3), we suggest the following flagging criteria:

○ SevEr: *if* $S_{M98} > 3$.

○ ModEr: *if* $2 < S_{M98} \leq 3$.

○ NoEr: *if* $S_{M98} \leq 2$.

**2.3   Outlier detection for half-hourly EC flux time series**

As described in Sect. 2.1, the second step in the data cleaning procedure consists in detecting outlying fluxes, and removing them if they inherited at least one ModEr statement. The outlier detection proceeds as follows: first, a signal extraction is performed, to obtain an estimate $\hat{y}_t^{\mathrm{true}}$ of the true signal; the residual term $\hat{e}_t$ is estimated as $\hat{e}_t = y_t^{\mathrm{obs}} - \hat{y}_t^{\mathrm{true}}$ and a PDF is built for it; outlying observations are then defined as those fluxes, for which the residual falls in the tails of the distribution, according
to pre-specified threshold values. In the general case, extreme values of model residuals can occur for three reasons: $i$) the model is misspecified and the residual is due to a wrong estimate for $\hat{y}_t^{\mathrm{true}}$; $ii$) the residual is due to the presence of a non-negligible systematic error (i.e. $|\beta_t| > 0$ in Eq. 2); or $iii$) the residual is indeed a tail instance on the PDF of $e_t$. Condition $i$) can be assessed by examining the degree of serial correlation in the residual time series. When a model is correctly specified, residuals should not show any serial correlation structure and resemble a white noise process. Distinguishing between conditions $ii$)
and $iii$) is not trivial, and would require an in depth analysis of the causes generating the anomalies. We propose to consider the presence of at least a ModEr statement as a symptom of condition $ii$) and thus to reject flux data identified as outliers and flagged with at least a ModEr statement. Otherwise, if all tests return NoEr statements, the outlying data is retained irrespective of the magnitude of the anomaly. Details about the modeling framework and how to take into account the heteroskedastic behavior of the residual component are provided in the following.

**2.3.1   Signal extraction**

For the purpose of signal extraction, we considered the following multiplicative model:

$$y_t^{\mathrm{obs}} = D_t \times LRT_t \times SRT_t \times I_t \tag{8}$$

where $D_t$ represents the diurnal cycle, $LRT_t$ represents the long-run (e.g. annual) trend, $SRT_t$ is the short-run (e.g. the intra-day) temporal dynamics term and $I_t$ is the irregular or residual component, considered as an estimate $\hat{e}_t$ of $e_t$ in Eq. (1). The





choice of a multiplicative model is preferred when the amplitude of the cyclic component is proportional to the level (i.e. the long-run trend) of the time-series (see Hyndman and Athanasopoulos, 2018, Ch. 6). Such a dynamic is typical of those ecosystems characterized by diurnal cycles of fluxes which are more pronounced during growing than during dormant seasons.

The estimation of each component in Eq. (8) can be performed in a variety of ways. In this work we used the STL (Seasonal-Trend decomposition based on LOESS - locally estimated scatterplot smoothing) algorithm developed by Cleveland et al.

(1990) and implemented in the *stlplus* R software package (Hafen, 2010, 2019). We choose the STL algorithm because of its flexibility and computational efficiency in modeling both the cyclic and the trend components of any functional forms and because of its ability to handle missing values.

The main steps of the STL algorithm are as follows. To separate out the diurnal cycle component, STL fits a smoothing curve to each sub-series that consists of the points in the same phase of the cycles in the time series (i.e. all points at the same half-

hour of day, for diurnal cycles). After removing the diurnal cycle, it fits another smoothing curve to all the points consecutively to get the trend components. This steps can be executed iteratively in presence of outliers. In particular, the STL algorithm deals with outliers by down-weighting them and iterating the procedure of diurnal cycle and trend components estimation. As the diurnal cycle and trend are smoothed, outliers tend to aggregate in the irregular component. The span parameters of the loess functions used for the extraction of the $D_t$ and $LRT_T$ were set equal to 7 days, while those for $SRT_t$ was set equal to

7 half-hours. The degree of locally-fitted polynomials in $D_t$ and $LRT_t$ extraction were imposed to be linear, while those in $SRT_t$ extraction was imposed to be constant. The number of iterations involved was set equal to 20.

The ability of the model to adequately describe the dynamics of the time series under investigation is performed by analysing the statistical properties of the residual time series. As mentioned above, a properly fitted model must produce residuals that are approximately uncorrelated in time. Any pattern in residuals, in fact, indicates the fitted model is misspecified and,

consequently, some kind of bias in fitted values is introduced. Towards this end, spectral properties of the incomplete residuals are assessed through the Lomb-Scargle (LS) periodogram (Lomb, 1976; Scargle, 1982). This method is particularly suited to detect periodic components in time series that are unequally sampled as a consequence of the presence of missing data at this stage of the data cleaning procedure.

### 2.3.2   Outlying flux data

Previous studies focusing on flux random uncertainty quantification have evidenced an heteroskedastic behavior of the random error (see Richardson et al., 2012, and references therein). Furthermore, it has been demonstrated that the random error distribution can be better approximated by a Laplace than by a Gaussian distribution (Vitale et al., 2019). To take into account these features, residuals are first grouped into 10 clusters of equal size (defined by percentiles) based on the estimated flux values. Then, for each of the 10 clusters, the outlier detection is performed assuming a Laplace distribution and taking into account

the $(1 - \alpha)\%$ confidence interval with $\alpha = 0.01$ given by $\mu \pm log(\alpha)\sigma$, where $\mu$ (location) is estimated by the median and the scale parameter, $\sigma$, is estimated through the $Q_n$ estimator by Rousseeuw and Croux (1993) for the same reasons introduced in Section 2.2.1.



It is important to note here that a significant limitation of the proposed outlier detection method is the inability to detect systematic errors whose sources act constantly across a flux time series, for long periods of time. As an example, a miscali-

bration of the instruments that persists for days, would most likely not be identified as outlying fluxes by the proposed outlier detection procedure. Such sources of systematic errors should be prevented via appropriate QA actions or, at least, specific QC tests should be devised, able to mark those time-series with SevEr statements.

## 2.4 Monte Carlo simulations

This work has made extensive use of Monte Carlo experiments that make use of simulate time series mimicking raw EC

datasets. The main purpose of the simulations is to create pairs of reference time series with known covariance structure that, after being contaminated with a specific source of systematic error, allow a quantitative and objective evaluation of $(i)$ the bias effect the source of systematic errors have and $(ii)$ the ability of proposed tests to correctly detect them. We note that, to these purposes, it is not strictly required to simulate medium- or long-term time series with realistic joint probability distributions from which to generate half-hourly fluxes with typical diurnal and seasonal cycles. In fact, it is reasonable to assume that QC

tests exhibiting poor performances on simulated data, have little chance of success when applied to observed time series which can exhibit more complex structures either of the underlying signal and of the (random) measurement error process, and which can be contaminated by the simultaneous occurrence of several sources of systematic errors.

Based on the main properties of EC time series (summarized in SM-Appendix A), synthetic time series were created from two autoregressive processes of order 1 (hereafter denoted AR; see SM-Appendix B for an overview of their stochastic prop-

erties) representative of the vertical wind speed and of the atmospheric scalar. The procedure adopted to ensure that simulated AR processes have a pre-fixed correlation structure is described in SM-Appendix C. All simulations were executed in $R$ programming environment (R Development Core Team, 2019).

Several scenarios have been considered to simulate stationary time series with different degrees of temporal dependence, and with pre-fixed correlation structures in order to simulate fluxes of different magnitude. All simulated time series have 18000

data points as in EC raw high-frequency time series sampled at 10 Hz and collected in 30-minute files. Once simulated, time series were then contaminated with specific sources of systematic error, details of which will be provided in Section 3.

## 2.5 Eddy-covariance datasets and flux calculations

In this study, data from ten EC sites part of the ICOS network (www.icos-ri.eu) were used: BE-Lon (Lonzee, Belgium, cropland; Moureaux et al., 2006), CH-Dav (Davos, Confoederatio Helvetica, evergreen needleleaf forest; Zielis et al., 2014), DE-

HoH (Hohes Holz, Deutschland, alluvial forest; Wollschläger et al., 2017), DE-Rus (Selhausen Juelich, Deutschland, cropland; Mauder et al., 2013), FI-Hyy (Hyytiala, Finland, evergreen needleleaf forest; Suni et al., 2003), FI-Sii (Siikaneva, Finland, wet grassland; Haapanala et al., 2006), FR-Fon (Fontainebleau, France, deciduous broadleaf forest; Delpierre et al., 2015), IT-SR2 (San Rossore 2, Italy, evergreen needleleaf forest; Matteucci et al., 2015), SE-Htm (Hyltemossa, Sweden, evergreen needleleaf forest; van Meeningen et al., 2017), SE-Nor (Norunda, Sweden, boreal forest; Lagergren et al., 2005). Field data were used



only as examples of real data and not to evaluate the proposed scheme, which is instead evaluated based on synthetic time series.

The turbulent fluxes of $CO_2$ (FC, $\mu$mol $CO_2$ m$^{-2}$s$^{-1}$), sensible heat (H, Wm$^{-2}$) and latent heat (LE, Wm$^{-2}$) were calculated from the covariances between the respective scalar and the vertical wind speed ($w$, ms$^{-1}$) following the standard EC calculation method (see, e.g., Sabbatini et al., 2018). The Net Ecosystem Exchange of $CO_2$ (NEE, $\mu$mol $CO_2$ m$^{-2}$s$^{-1}$) was estimated by

integrating FC with the concurrent storage flux (SC, $\mu$mol $CO_2$ m$^{-2}$s$^{-1}$). The EddyPro® software (LI-COR Biosciences, 2019) was used to this aim, employing, the double coordinate rotations for tilt correction, 30 min block averaging, the maximum cross-covariance method for time lag determination and the in-situ spectral corrections proposed by Fratini et al. (2012).

## 3  Results and discussion

### 3.1  Performance evaluation of QC tests

#### 3.1.1  Low signal resolution (LSR) test

A simulation study was carried out with the twofold aim of quantifying the bias caused by low signal resolution problems and evaluating the performance of the LSR test described in Section 2.2.1. To this end, error-free AR processes of length $n = 18000$ were first simulated and subsequently contaminated with artificially generated errors. The correlation between time series has been set to 0.1, simulating medium-to-low fluxes, while the autoregressive coefficient ($\phi$) was allowed to take on a

value in the set [0.90, 0.95, 0.99] to represent different degrees of serial dependence (i.e. autocorrelation) as observed in EC data (SM-Appendix A). Low resolution problems were simulated $i)$ by rounding the simulated values from 2 to 0 digits, $ii)$ by sampling at random an increasing percentage of data (15, 30, 45, 60, and 75% of the sample size) and then replacing it via the Last Observation Carried Forward (LOCF) imputation technique. This allows to create the typical ramp structure commonly encountered in raw, high-frequency time series. Each of the 6 scenarios (3 values of $\phi \times 2$ types of error, abbreviated $S_{i=1,...,6}^R$)

was ran 199 times to obtain robust statistics. Three realizations of contaminated AR processes are depicted in Figure 2.

Bias affecting correlation estimates was quantified as the difference in absolute percentage between the correlation estimated on error-free ($\rho_{EF}$) and on contaminated ($\rho_C$) data. Results for each scenario are shown in Figure 3. When low resolution problems were simulated by rounding values ($S_{i=1,2,3}^R$), the bias was less than 2%. Similar results were reported by VM97 and more recently by Foken et al. (2019). On the contrary, when low resolution problems were simulated by replacing values via LOCF,

a significant bias was introduced in correlation estimates, whose amount depends not only on the number of contaminated data, but also on the degree of serial dependence. As expected, the percentage of data affected by error being equal, the lower the degree of serial correlation, the higher the amount of bias. In $S_4^R$, when the AR processes were simulated with $\phi = 0.90$, a bias greater than 5% was observed as soon as more than 30% of data were contaminated by error. On the contrary, when $\phi = 0.99$ as in $S_6^R$, the bias was less than 5%, even when 75% of data were contaminated by error. This is because when a series has

strong autocorrelation, following values are expected to be close to current value and therefore, replacing them with the current value does not change the time dynamic significantly and as a consequence the correlation estimate remains unbiased.

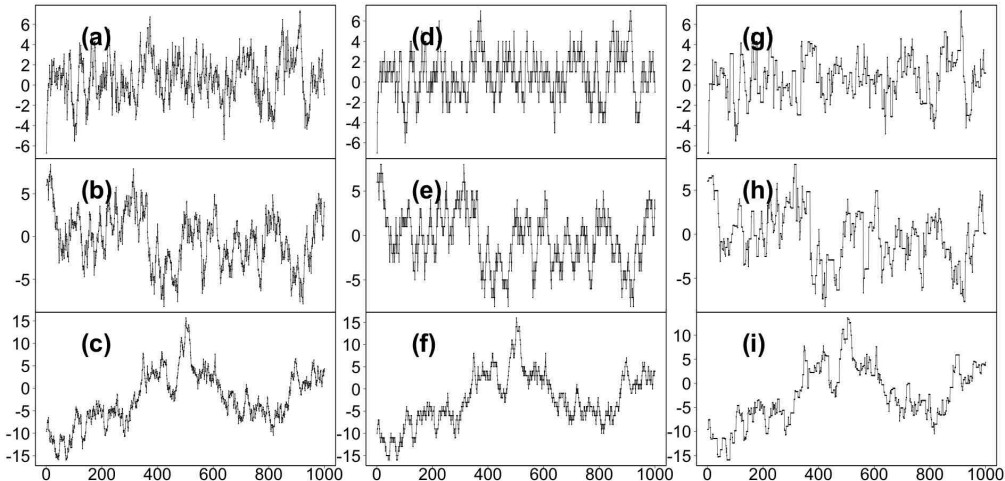

**Figure 2.** Three realizations of AR processes of length 1000 with $\phi = 0.9$ (panel a), $\phi = 0.95$ (panel b) and $\phi = 0.99$ (panel c), affected by low resolution problems simulated by discretizing the original variable (panels d-f) and by replacing 60% of original data via Last Observation Carried Forward (LOCF) technique (panels g-i).

The ability of the LSR test to disentangle among these situations can be appreciated by looking at the distribution of the $R^2$ values in the 6 scenarios as shown in Figure 3 (middle panels). To aid in comparison we also added the results of the amplitude resolution test by VM97 (hereafter A-R, right panels of Figure 3). Considering threshold values at 0.99 and 0.995 to identify

SevEr and ModEr statements, the LSR test showed a good performance with much lower false positive and false negative rates lower than A-R test. For the latter, low-resolution problems were identified only for scenarios $S_{i=1,2,3}^R$ and limited to cases when the number of digits is reduced to zero. The amount of bias in such cases was however negligible. On the contrary, the sensitivity of the LSR test was such that the higher the bias affecting correlation estimates the lower the $R^2$ values, irrespective of the low-resolution simulation method.

We applied both LSR and A-R testing procedures to the selected field datasets described in Section 2.5 (results are reported in SM-Appendix D). For LE and NEE, similar amounts of data were flagged by the two tests. On the contrary, for sensible heat flux the LSR test flagged much less data than the A-R test. In particular, for 5 sites the percentage of H data hard-flagged by the A-R test exceeded 40% with a maximum of 68.3% at SE-Htm. The hard-flag was often assigned by the A-R test to the sonic temperature time series during periods of low variability, which led to the typical step ladder in the data. However, in most of

these cases, that does not necessarily imply biased covariance estimates.

To aid in comparison, some illustrative examples are shown in Figure 4. Panel a) shows a sonic temperature time series for which both tests provide negligible evidences of error. Notice that in this case, the $R^2$ is close to unity although 19.8% of data was constituted by repeated values. The case shown in panel b) is representative of contrasting results: the A-R test assigned a hard-flag, while the LSR test returned a NoEr statement. By visual inspection, it would seem that despite the step-ladder

appearance in the data, the time series dynamic is mostly preserved. In these occurrences, bias affecting covariance estimates

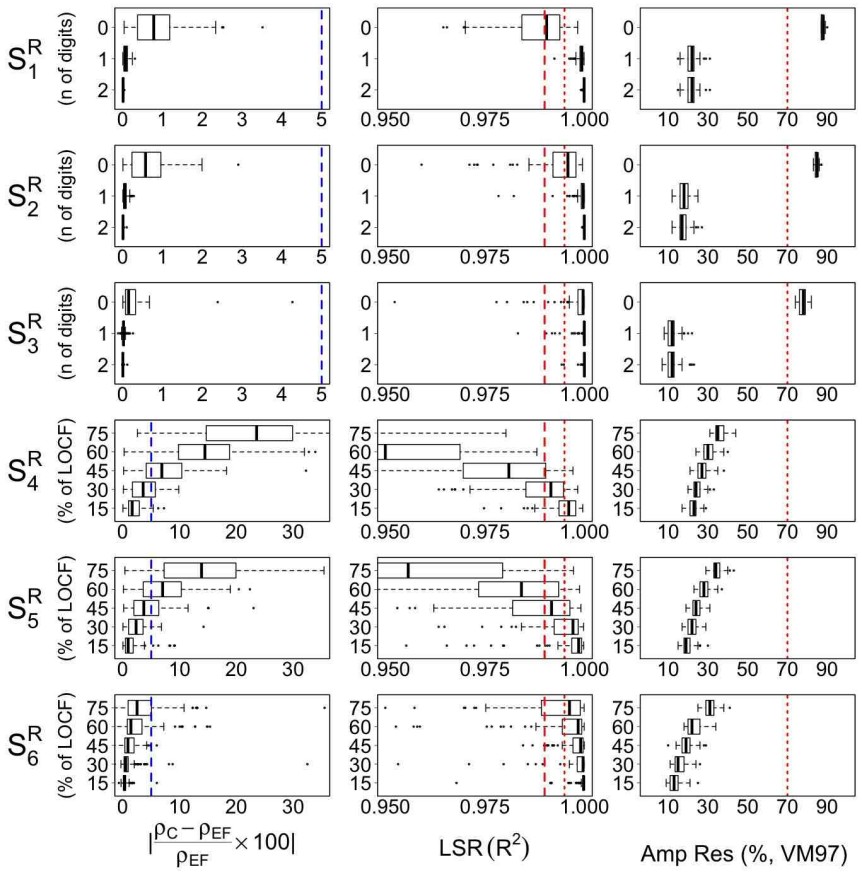

**Figure 3.** Bias effect of low signal resolution problems and test performance evaluation. Left panels: difference in percentage between correlation estimated on error-free time series ($\rho_{EF}$) and on data contaminated ($\rho_C$) by low resolution problems (dashed blue line indicates 5% bias). Middle panels: distribution of $R^2$ values for the LSR test; dashed red lines indicate the threshold values at 0.99 and 0.995 adopted for defining the SevEr and ModEr statements, respectively. Right panels: distribution of the A-R test statistic by VM97; dashed red lines indicate the threshold value at 70% as recommended by the authors for assign a hard-flag to data. In $S_1^R$ and $S_4^R$, $\phi = 0.90$; in $S_2^R$ and $S_5^R$, $\phi = 0.95$; in $S_3^R$ and $S_6^R$, $\phi = 0.99$.

is typically negligible, as demonstrated by the fact that the CCFs estimated with original data and after removal of repeated values overlap almost perfectly. The cases depicted in panels c) and d) are representative of situations of strong resolution issues leading to diverging CCFs and significant biases. The SLR test successfully detects both problems, while A-R test fails to flag case c).





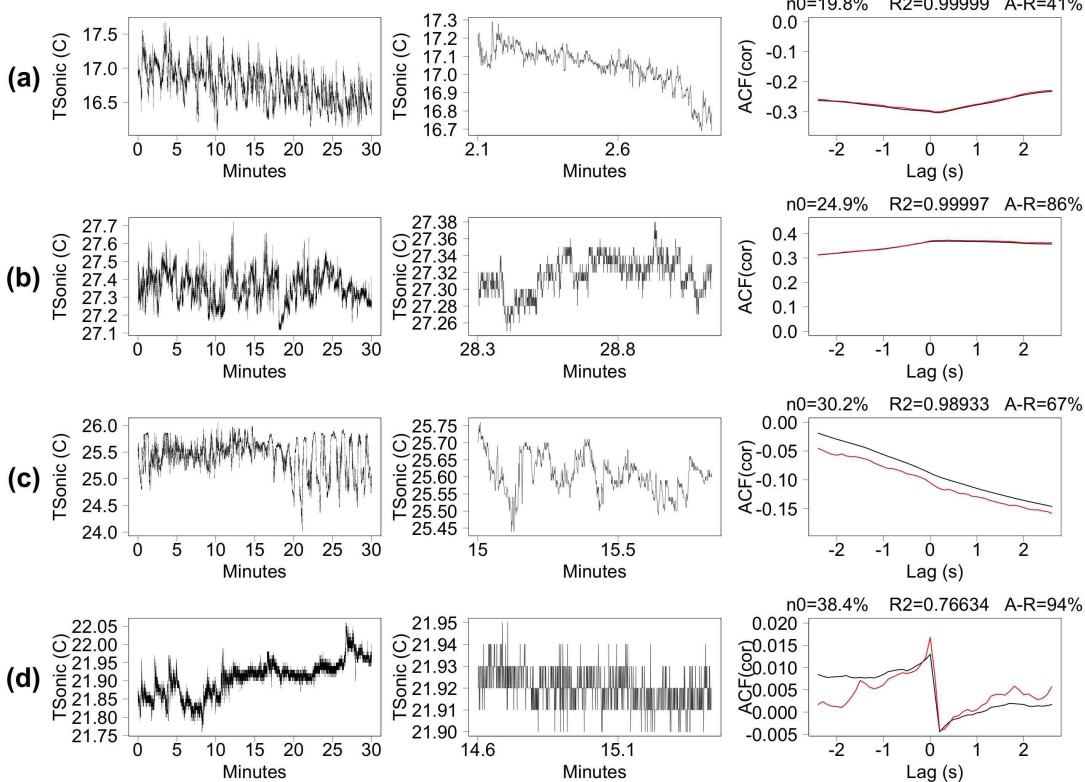

**Figure 4.** Comparison of LSR test and amplitude resolution (A-R) test by VM97 on observed data collected at SE-Nor site. Sonic temperature time series collected in 30 minutes (left panels) and in a shorter temporal window of length equal to 1000 timesteps (middle panels). Right panels: cross-correlations between $w$-scalar time series with original data (black line) and after removal of repeated values (red line); on the top: $n0$ indicate the percentage of repeated consecutive values; $R^2$ and A-R denote the statistics of the low signal resolution tests.

## 520 3.1.2 Structural changes tests

The Monte Carlo experiment designed to the evaluation of the performance of the proposed structural changes tests described in Section 2.2.1 involved 6 scenarios, $S_{i=1,..,6}^{SC}$, where several synthetic AR processes ($\phi = 0.99$) of length $n = 18000$ were contaminated by:

$S_1^{SC}$ : a stochastic trend;

$S_2^{SC}$ : a deterministic linear trend;

$S_3^{SC}$ : an abrupt change in the mean level whose duration and shift were fixed at 3000 timesteps and 3 times the interquartile distance (IQD) of the data, respectively;

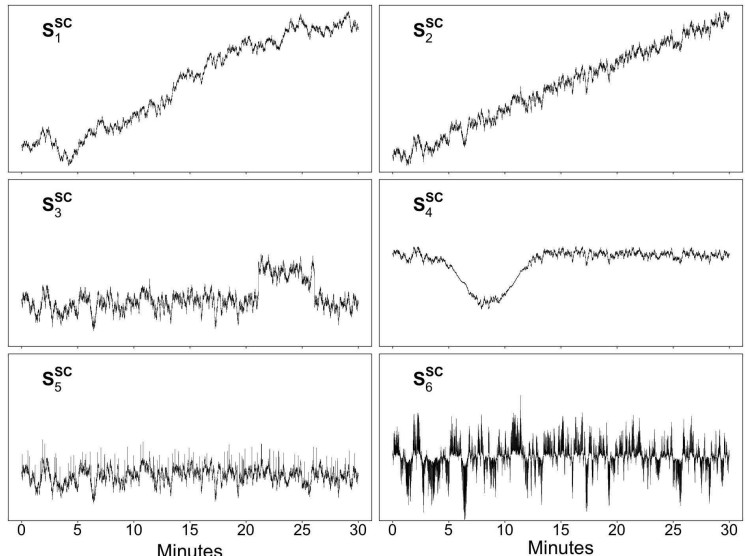

**Figure 5.** Illustrative example of an AR process contaminated by structural changes according to $S^{SC}_{i=1,\dots,6}$ scenarios.

$S^{SC}_4$ : multiplying a block of consecutive data of size 6000 by a cosine function to mimic episodic burst events as often observed in real data;

$S^{SC}_5$ : introducing 0.5% of spiky data generated by adding $2\times$IQD to the original data;

$S^{SC}_6$ : replacing 15% of the data with the original values multiplied by 5, to simulate changes in variance.

Although these scenarios only cover a fraction of the problems encountered in real observations, the experiment aims at evaluating the test sensitivity in presence of aberrant structural changes which, in most cases, can only be imputable to the malfunctioning of the measurement system. Note that scenarios $S^{SC}_{i=1,2}$ do not aim at simulating time series affected by structural

changes. Rather, their purpose is to evaluate the propensity of the tests to false positive errors.

Illustrative examples of simulated time series are shown in Figure 5. Each scenario was ran 199 times. For each simulation, the statistics of the $HF_5$, $HD_5$ and KID tests were calculated. As a reference, we also applied the "discontinuity" and "kurtosis index" tests proposed by VM97 (while their "dropouts" and "skewness" tests resulted insensitive to the simulated scenarios and will therefore not be further discussed). Results are summarized in Figure 6.

We observe that all tests exhibited a low false positive error rate (cfr. scenarios $S^{SC}_1$ and $S^{SC}_2$). The only exception was the VM97 test based on the Haar transform to detect discontinuities in the mean level, which instead showed a low performance when time series was contaminated by a stochastic trend component. The sudden shift in mean level simulated in $S^{SC}_3$ was correctly identified by the KID test in most simulations. The Haar transform for detecting discontinuity in the mean level identified the simulated structural changes, but only in less than 50% of cases the hard-flag was assigned. Good performances





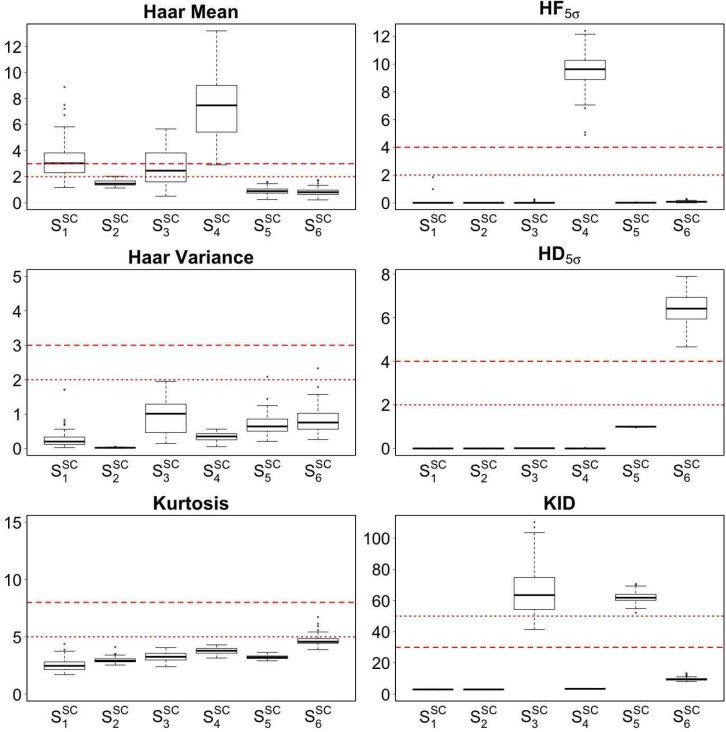

**Figure 6.** Distribution of results for the structural changes detection tests, in six different scenarios. Short and long dashed red lines indicate the threshold values adopted for defining SevEr and ModEr statements (*hard-* and *soft-flag* for VM97 tests), respectively.

were observed in $S_4^{SC}$ for both the discontinuity in mean (VM97) and $HF_5$ tests. Structural changes caused by heteroskedastic behavior simulated in $S_5^{SC}$ and $S_6^{SC}$ were correctly detected by only KID and $HD_5$ tests, respectively.

The application of the testing procedures on actual EC time series have shown a higher sensitivity of the VM97 tests, compared to the newly proposed tests, with a tendency to assign hard-flags even in cases no evidence of instrumental error was supported by visual inspection. On the contrary, the proposed tests resulted more selective at identifying data affected by

structural changes, although in some case such structural changes were not necessarily imputable to instrument malfunction. In most of these occurrences, however, structural changes are indicative of non-stationary conditions, which as we know is another source of systematic error that introduces bias in covariance estimates (see Section 3.1.3). This is an example where two tests are not fully independent and could identify the same issue, reason why the ModEr statements are not combined. Illustrative examples of application of the testing procedures on raw data are shown in Figure 7.

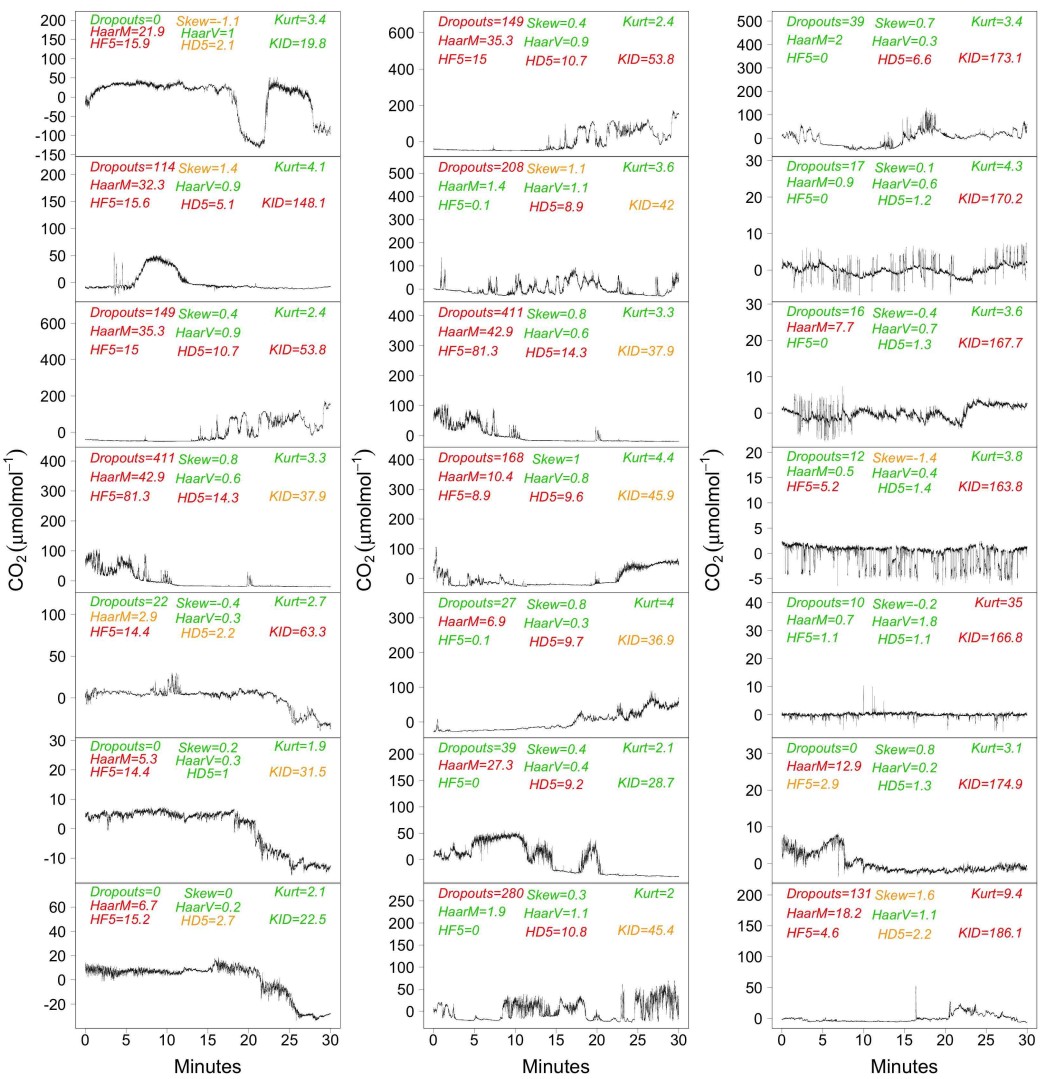

**Figure 7.** Application of tests for structural changes detection on a selection of $CO_2$ time series ($\mu$molmol$^{-1}$, after mean removal) collected at BE-Lon site. The statistics of dropouts, skewness, kurtosis, discontinuities in mean (HaarM) and variance (HaarV) tests as described by VM97, and of HF$_5$, HD$_5$ and KID test described in Section 2.2.1 are reported on the top of each panel. For each test, green text indicates NoEr (no flag by VM97 tests), orange ModEr (*soft* flag), red text SevEr (*hard* flag).

**555** **3.1.3 Stationarity tests**

We compared the performance of the stationary tests by FW96 and by M98 via Monte Carlo simulation making use of synthetic bivariate pairs of AR processes, $x_t$ as representative of vertical wind speed and $y_t$ as representative of scalar atmospheric concentrations, with $t = 1, ..., n = 18000$, according to the following scenarios:





$S_1^S$ : $\phi_x, \phi_y = 0.95$ and $\rho(x,y) = 0.05$ simulating fluxes of low magnitude.

$S_2^S$ : $\phi_x, \phi_y = 0.95$ and $\rho(x,y) = 0.25$ simulating fluxes of high magnitude.

$S_3^S$ : $\phi_x, \phi_y = 0.99$ and $\rho(x,y) = 0.05$ simulating fluxes of low magnitude, but in presence of time series with high degree of serial correlation.

$S_4^S$ : $\phi_x, \phi_y = 0.99$ and $\rho(x,y) = 0.25$ simulating fluxes of high magnitude, but in presence of time series with high degree of serial correlation.

$S_5^S$ : as in $S_3^S$ , where both $x_t$ and $y_t$ were contaminated by deterministic linear trend components.

$S_6^S$ : as in $S_4^S$ , where both $x_t$ and $y_t$ were contaminated by deterministic linear trend components.

$S_7^S$ : as in $S_1^S$ , where both $x_t$ and $y_t$ were contaminated by stochastic trend components.

$S_8^S$ : as in $S_2^S$ , where both $x_t$ and $y_t$ were contaminated by stochastic trend components.

Notice that $S_{i=1,..,4}^S$ simulate fluxes measured under stationary conditions (by definition, since $|\phi_x, \phi_y| < 1$), while the
remaining $S_{i=5,...,8}^S$ simulate fluxes measured under non-stationary conditions. To mimic trend dynamics of magnitude similar to those observed in real cases, the slope of the deterministic linear trend function $f(t) = \beta \cdot t$, was fixed equal to 0.0004 and 0.004 for $x_t$ and $y_t$, respectively, whereas the stochastic trend components were generated as $\sum_{t=1}^{n} \varepsilon_t$, where $\varepsilon$ is drawn from a Normal distribution with mean 0 and standard deviation equal to 0.025 and 0.25 for $x_t$ and $y_t$, respectively. Exemplary realization of the simulated AR processes and their CCFs estimated on either single run and averaged over multiple runs (i.e.
ensemble CCF) are illustrated in Figure 8.

Results of applying the FW96 and M98 stationarity tests to 100 simulation runs for each of the 8 scenarios considered are shown in Figure 9. Test performances were evaluated through a statistical sensitivity analysis given by the percentage of correctly identified cases. The threshold values used to assign ModEr and SevEr statements were 30% and 100% for the FW96, and 2 and 3 for the M98, respectively.

In scenarios $S_1^S$ and $S_2^S$, where simulated time series are stationary and characterized by a lower degree of serial correlation, both tests exhibited good performances. The fraction of cases in which the statistics exceeded the threshold values that would, wrongly, lead to the rejection of data (i.e. 100% for FW96 and 3 for M98) was less than 5%, with a slightly better score for M98. In $S_3^S$ and $S_4^S$, with stationary time series characterized by a higher degree of serial correlation, the performance of the FW96 test appeared discordant: unlike the excellent performance obtained in $S_4^S$, in $S_3^S$ the test statistic was higher than 30%
in 44% of cases, and, greater than 100% in another 10%. The cause of this low performance is related to the sensitivity of the FW96 test to cases in which its denominator (i.e. covariance over the entire period) approaches zero and thus tends to make the ratio diverge, independently from the quantity at the numerator. The M98 test showed instead a lower sensitivity to low correlations among variables, but a greater sensitivity to the degree of serial correlation. In both $S_3^S$ and $S_4^S$, the percentage of data in which the test statistic was greater than 2 was in fact higher than 20%. At the same time, however, in only 1% of cases

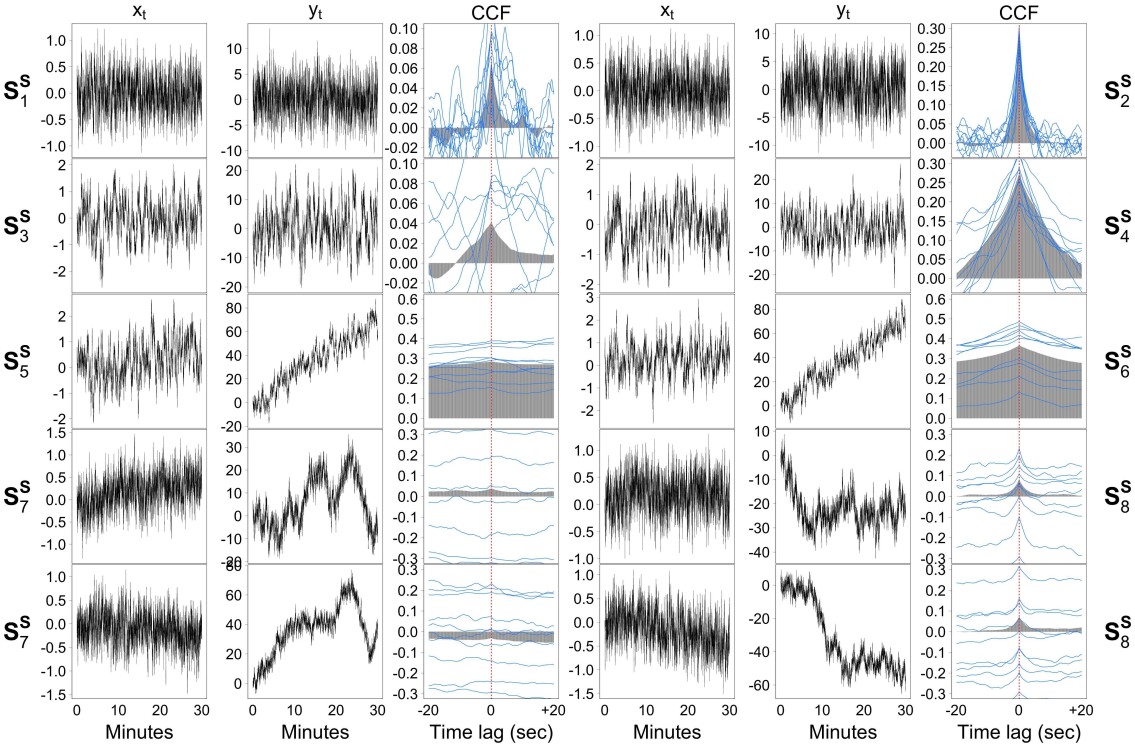

**Figure 8.** Illustrative examples of simulated $x_t$ and $y_t$ AR processes and their cross-correlation function (CCF) in each of the 8 scenarios designed to evaluate the performance of the stationarity tests. Grey areas represent the ensemble CCF averaged over multiple simulation runs. Blue lines represent CCF estimated in 10 individual simulation runs.

the statistic exceeded 3, the threshold value adopted to assign a SevEr statement. This result indicates that the use of 3 instead of 2 as a threshold value for the M98 test is preferable since it reduces the false positive error rate.

In both $S_5^S$ and $S_6^S$, the M98 test showed an excellent ability at detecting non-stationarity caused by the presence of deterministic linear trend components: the statistic was higher than 3 in 95% of cases. On the contrary, for the FW96 test a similar performance would require the use of a 30% threshold value while according to the recommendations by Foken et al. (2004),

when the value of the statistics is between 30% and 100%, if well-developed turbulence conditions are satisfied, the data would not be rejected, but classified as data of intermediate quality. In the presence of stochastic trend components as simulated in scenarios $S_7^S$ and $S_8^S$, the M98 test performed better than the FW96 test. However, the false negative error rate (i.e. data erroneously considered stationary) remained higher than 20% when using 3 as a threshold value. In particular, only 79% and 67% in the $S_7^S$ and $S_8^S$, respectively, received the status of data affected by SevEr.

The performance considerations described above are confirmed when tests are applied to observed data. Some examples are shown in Figure 10. Time series represented in panels a-c refer to cases in which both tests provided strong evidence of non-stationarity ($S_{FW96} > 100\%$, $S_{M98} > 3$). In these cases, the difference between the average of 5-min covariances and

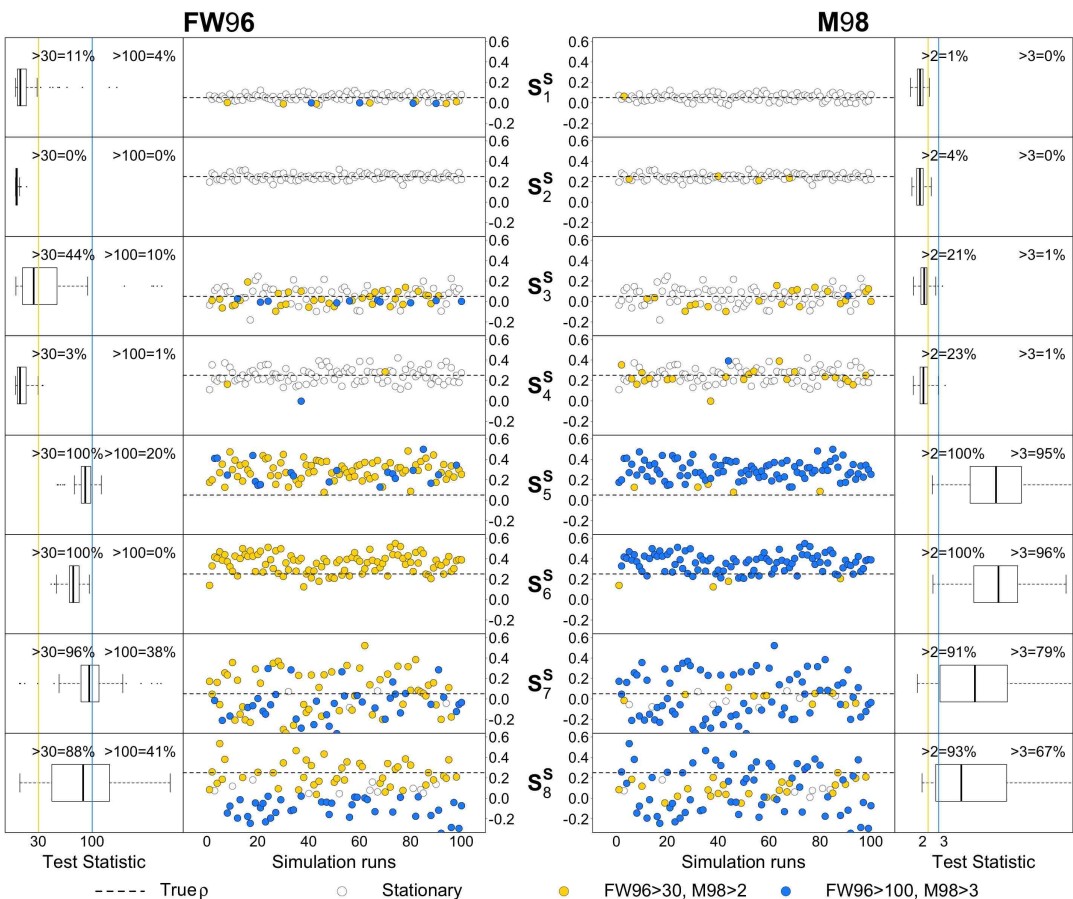

**Figure 9.** Performance evaluation of stationarity tests by Foken and Wichura (1996, FW96) and by Mahrt (1998, M98) in each of the 8 simulated scenarios. Left and right panels: distribution of the test statistic; blue and yellow lines indicating threshold values. Middle panels: distribution of the test statistic as a function of the correlation between variables in 100 simulation runs; blue and yellow points indicating simulations when the test statistic exceeds the corresponding threshold values (percentage values are reported at the top of left and right panels).

those estimated over 30-min is indeed significantly different from zero (right-hand panels). Panels d-f represent 3 situations in which the M98 test provided evidence of non-stationarity ($S_{M98} > 3$) while the FW96 test returned high- or intermediate quality assessments, as the difference between the average of 5-min covariances and those estimated over 30-min is negligible. However, in such cases 5-min covariances are obviously affected by considerable variability and/or trends, and only happen to provide a mean value close to the 30-min covariance. Therefore, we suggest that the FW96 test provides a necessary, but not sufficient, condition for stationarity. Conversely, the M98 test provides a satisfying performance also in such cases. Examples in panels g-i depict 3 situations in which the FW96 test provided strong evidence of non-stationarity ($S_{FW96} > 100\%$) while the M98 test provided, at most, weak evidence (NoEr or ModEr). As mentioned earlier, such disagreement often only occurred



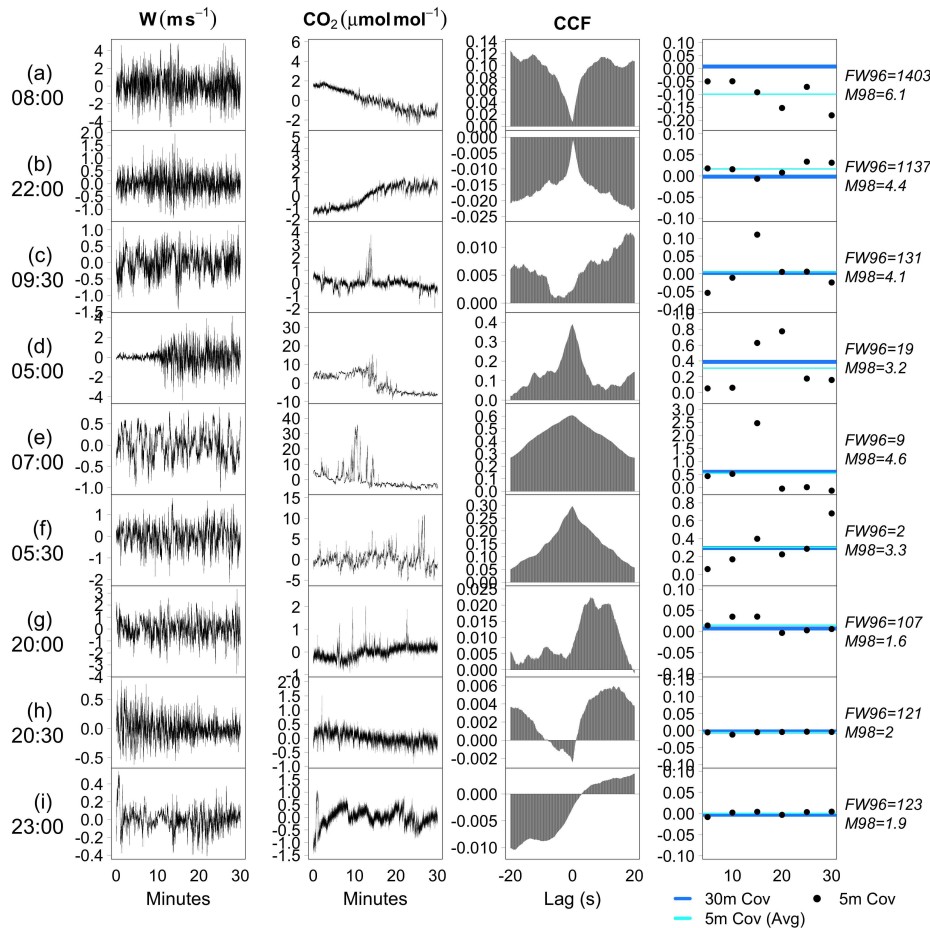

**Figure 10.** Application of stationary tests on a selection of EC raw data collected at FI-Hyy site. From left to right: vertical wind speed ($w$, ms$^{-1}$); $CO_2$ time series ($\mu$molmol$^{-1}$, after mean removal); cross-covariance function (CCF); comparison of 5-min covariances (black points), their average (cyan lines) and 30-min covariance (blue lines). FW96 and M98 test statistics are reported on the right-hand side.

on account of the 30-min covariance being close to zero. As can be observed for these selected cases, in fact, not only the differences between the average of 5-min covariances and those estimated on the whole period of 30-min is negligible, but also the degree of dispersion of the individual 5-min covariances remain at low levels, as it is expected in stationary conditions.

## 3.2    Application of the data cleaning procedure

In this Section we report the results of the data cleaning procedure based on the workflow depicted in Figure 1 and including the application of QC tests described in Section 2. An illustrative example for NEE time series collected at FI-Sii site during a period of 3 months is shown in Figure 11 (for the other flux variables and for all the other sites, we refer the reader to the SM-Appendix D). The original time series had a rate of missing data of 12.3%. The data cleaning procedure eliminated


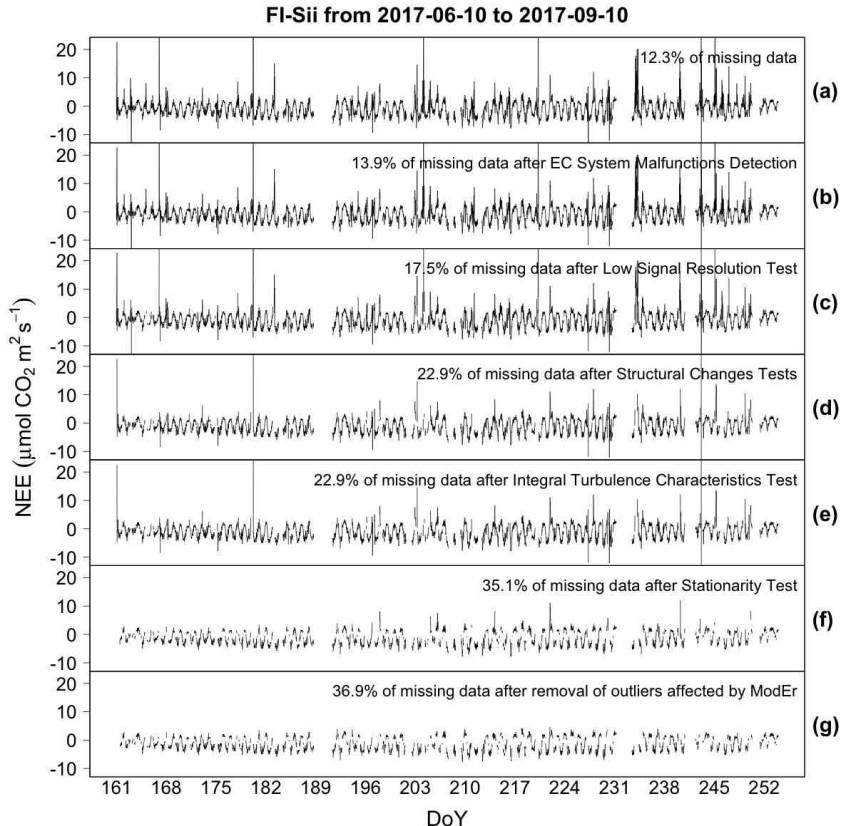

**Figure 11.** Illustrative example of the sequential data cleaning procedure applied to NEE fluxes at FI-Sii site.

another 24.6% of the data, for a total of 36.9% resulting data gaps. In particular, 1.6% of data was rejected after the removal
of unreliable data caused by system malfunction and disturbances (panel b); 3.6% was removed due to low signal resolution
problems (panel c); 5.4% was eliminated due to evidence of aberrant structural changes (panel d); the ITC test did not remove
any data (panel e); an additional 10.6% of data were removed because of nonstationary conditions (M98 test, panel f); finally,
1.6% of fluxes were rejected because identified as both outlying and inheriting at least one ModEr (panel g).

Figure 12 reports the percentages, averaged over the 10 sites under investigation, of H, LE and NEE flux data for which the
QC tests returned ModEr and SevEr statements. The highest percentage of data inheriting a SevEr statement is caused by the
lack of stationary conditions: irrespective of the flux variable, the percentage was around 20%, with a maximum peak of 36.9%
for LE fluxes at CH-Dav site and a minimum of 10.2% for H fluxes at SE-Htm site (see SM-Appendix D). With respect to the
non-stationary test by M98, a further 25% of fluxes inherited a ModEr statement. The percentage of data where the ITC test
returned SevEr (ITC> 100%) was only 0.1%, while about 6% received the status of ModEr.

Severe low signal resolution issues were identified only sporadically (in no more than 2.4% of flux values), while a ModEr
statement for $HD_B$ test was inherited by 18% of H values, due also to a moderate heteroskedasticity often observed in differ-



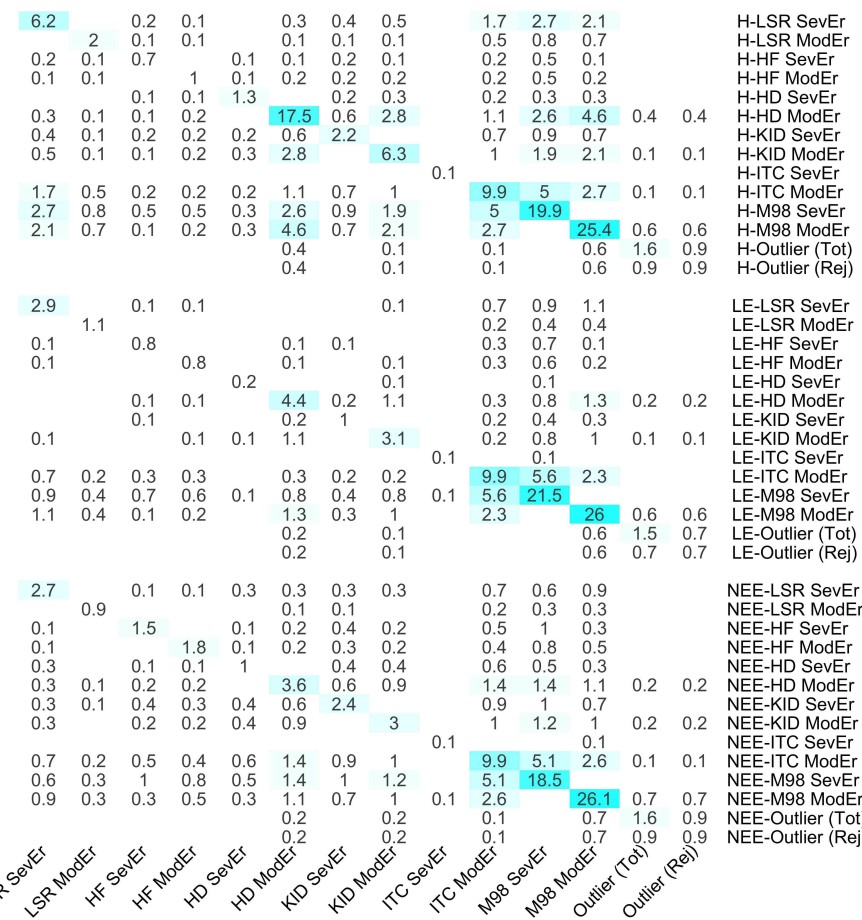

**Figure 12.** Percentages of H, LE and NEE flux data affected by specific sources of systematic error according to several QC tests. Each cell of the matrix indicates the percentage of data receiving the status indicated in the $i^{th}$ row and the $j^{th}$ column; those on the diagonal refers to the percentage of data identified by each individual test. Empty cells indicate no data identified. Values are averaged across the 10 EC sites under investigation.

enced sonic temperature time series. For all 3 flux variables, the percentage of outlying data was around 1.5%. Approximately half of them received at least a ModEr statement from one of the QC tests and were therefore removed.

Figure 13 shows the STL decomposition (Cleveland et al., 1990) of NEE time series at FI-Sii site using the set of parameters
described in Sect. 2.3. The ability of the modelling approach to separate the signal from the noise was assessed by evaluating the spectral characteristics of the irregular component by means of the Lomb-Scargle (LS) periodogram (Lomb, 1976; Scargle, 1982). In general, the periodogram did not show significant peaks, as exemplified for NEE in Figure 14 (panel g) and shown at length for all flux variables in the SM-Appendix D. Despite the ability of the STL algorithm to reliably reproduce the


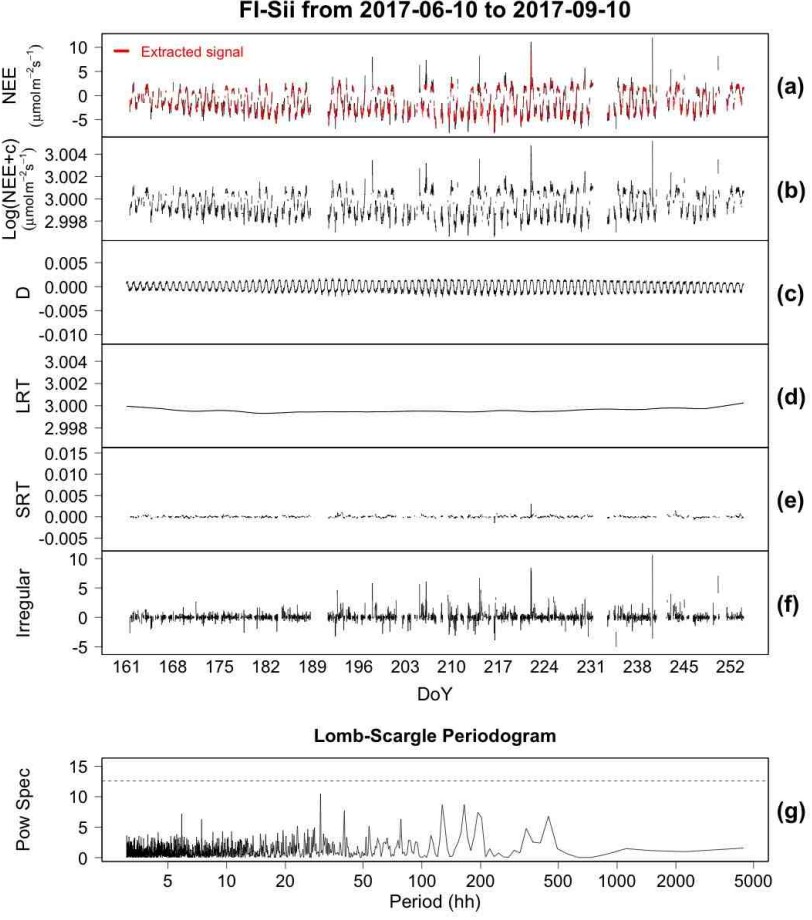

**Figure 13.** Example of STL decomposition applied to NEE time series collected at FI-Sii site. Panel g shows the Lomb-Scargle power spectral density estimate applied to the irregular component (panel f) of STL decomposition. Dashed line indicates the 0.01 significance level.

complex correlation structure present in EC flux data, the irregular component was often heteroskedastic. This means that even

if independent (since most of the serial correlation structure is removed), the irregular component is not identically distributed, i.e. its PDF changes over time. Such a property strongly limits the ability to use global threshold values above and below which data are identified as outliers.

Previous researches (e.g. Richardson et al., 2012; Vitale et al., 2019), have highlighted that the random error scales with the flux magnitude and that a Laplace distribution can better approximate the PDF of the random error. With this in mind,

we grouped the values of the irregular component into 10 clusters of equal size, defined by percentiles of the signal estimated by the STL. This way, each cluster should contain values of the irregular component that are more likely to be identically distributed. Assuming a Laplace distribution and the $(1 - \alpha)\%$ confidence interval at $\alpha = 0.01$ significance level, any values



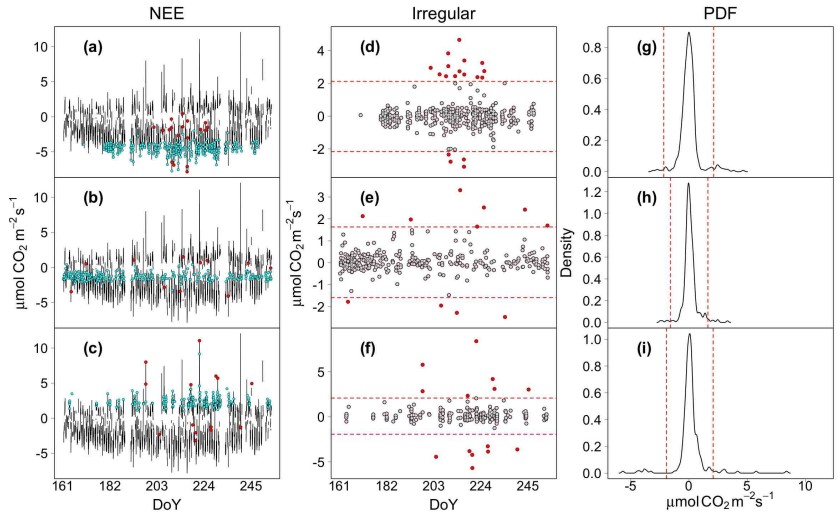

**Figure 14.** Illustrative example of the outlier detection procedure applied to NEE fluxes at FI-Sii site. NEE time series for the whole period (black lines) and selected flux values according to 3 deciles of the extracted signal (coloured points) are shown in panels a-c. The corresponding values of the irregular component are shown in panels d-f, while their probability density functions are depicted in panels g-i. Dashed red lines indicate the $\alpha$-outlier regions ($\alpha = 0.01$) assuming a Laplace distribution. Red points indicate the detected outliers.

exceeding $\pm 4.6 \cdot \sigma$ was detected as outlier (where $\sigma$ was estimated using the $Q_n$ estimator of Rousseeuw and Croux, 1993). An illustrative example of this procedure is depicted in Figure 14, where for simplicity the detail of only 3 clusters is reported.

## 4 Conclusions

Quality control of eddy covariance flux datasets is challenging. The sources of systematic error responsible for introducing significant biases in the flux computation are manifold, and their correct identification is often made difficult by the masking effect induced by both the intrinsic stochastic properties (e.g. high degree of serial dependence, heteroskedasticity) and by the high level of noise characterizing raw data.

To take into account these features, new tests have been developed and included in a robust data cleaning procedure where the data rejection is articulated in two stages: the first stage involves the removal of any flux data for which at least one of the QC tests returned strong evidences about the presence of a specific source of systematic error (SevEr); the second stage consists in the removal of outlying fluxes, provided that at least one of the QC tests returned weak evidences about the presence of sources of systematic error (ModEr) for the same flux value. Any flux data where all QC tests provided only negligible evidences of systematic error (NoEr) is never removed, even if it is later identified as an outlier in the flux time series.

Compared to the existing classification schemes, the proposed approach does not aim at assigning a quality flag to flux data by combining the results of different QC tests. Rather, it aims at ensuring its scalability in order to facilitate the inclusion of new tests beyond those proposed in this paper.



Although there is a strict relationship between the value of test statistics and the amount of bias affecting individual flux data,
the interpretation of SevEr, ModEr and NoEr statements is performed in probabilistic terms, as the chances the presence of a
source of systematic error is responsible for introducing bias in flux estimation. Consequently, the choice of threshold values
used to assign the SevEr, ModEr and NoEr statements are to be interpreted as indicative of the margin of error associated with
the result of a statistical test.

In this study, the performance evaluation of each proposed test was carried out mainly by means of Monte Carlo experiments.
Although the proposed scenarios are representative only of a part of the innumerable cases that can be encountered in real
applications, simulations offer undoubted advantages because they allow a quantitative and objective evaluation of the bias
effect each source of systematic error has on flux estimation, and of the ability of proposed tests to correctly detect them. The
design of increasingly complex scenarios and the building of a reference EC datasets are targets of ongoing work.

As for assigning a quality level to the retained data, we maintain that the estimates of random uncertainty is the appropriate
metric. In fact, as a general principle, the larger the random uncertainty, the larger the amount of measurement error and,
consequently, the lower the quality of the data. Assuming that flux data affected by systematic error have been avoided or
removed via appropriate QA/QC procedures, the use of random uncertainty estimates as a quality indicator 1) would not
constrain the QC test development; 2) would not preclude a classification of the data quality, if needed, and 3) would meet the
requirements of advanced methods of analyses where interval estimates are more important than individual point estimates,
such as in studies based on data assimilation techniques.

Although there is still room for improvement, particularly in the development of more performing QC tests aiming at detect-
ing violations of the main assumptions underlying the EC technique, we believe that the proposed data cleaning procedure can
serve as a basis toward a unified QC strategy suitable for the centralized data processing pipelines, where the use of completely
data-driven procedures that guarantee objectivity and reproducibility of the results constitutes an essential prerequisite.

*Author contributions.* DV and DP conceived the study. DV organized the structure, selected, proposed and implemented the methodologies
and did all the simulations, discussing the results with the coauthors. DV and GF wrote all the sections with the supervision of DP and with
the contribution of all the coauthors. All authors reviewed the final manuscript, approved it, and agreed for the submission.

*Competing interests.* The authors declare that they have no conflict of interest.

*Acknowledgements.* DV thanks the ENVRIPLUS H2020 European project (Grant Agreement 654182) for the support. GN thanks the
RINGO H2020 European project (Grant Agreement 730944) for the support. DP thanks the ENVRIFAIR H2020 European project (Grant
Agreement 824068) for the support. Data used are from the Integrated Carbon Observation System (ICOS) European Research Infrastructure
and accessible through the ICOS Carbon Portal (www.icos-cp.eu).



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
