# Peer review of "A robust data cleaning procedure for eddy covariance flux measurements"

_Biogeosciences, 2019_

## Referee Comment (RC1) · Andrew Kowalski (Referee) · 6 Aug 2019

The authors present a procedure for the "cleaning" of raw data from eddy covariance flux towers. If effective, such a procedure will be of great value. The procedure described is founded partially in previous studies of quality control for flux tower data, and partially based on statistical analyses that can be considered novel in this context. I am no expert in statistical analyses, but the approach put forth seems to be valid and defensible. The paper is well written, although I offer some minor suggestions intended for its improvement. My main problem with the paper is that the evaluation of the proposed scheme is based on synthetic time series, and I am left wondering whether such "data" are representative in terms of the types of problems that can be encountered in real, field data. If there were a means of assessing this key question,

then I would suggest this as a "major" revision issue. In any event, however, I believe that the manuscript merits publication.

Specific comments (by line number)

38 "An intergral part of the EC..."

82 This statement strikes is as excessively bold and over-simplified. For example, a random error that were to consistently reduce the covariance at a frequency of 0.01Hz (by introducing random noise at that frequency) would cause flux underestimation for the corresponding eddies. It could well be that these frequencies matter greatly for (convective) eddy transport during daytime, and far less so at night. The result would be an overestimation of NEE over the long term.

83 "classified into partially versus completely data-driven"

87 "Such a call"

93 sacrifice

98 "because it is extremely"

100 "false-positive rates"

105 "selected, the results of these tests"

135 "proposed set, and iv) results in"

142 "a set of tests"

200 "with some instruments, especially older models or often"

200 Table 1: if the terms "fraction of missing records (FMR)" and "longest gap duration (LGD)" are defined with the acronyms in parentheses as in this sentence, then it makes it easier for the reader to find them when they encounter them for the first time in the text.

248 Again, I find greatly exaggerated the claim the random errors due not affect covariances. This depends on the magnitudes of the signal/noise variances, and also on their durations during the averaging period.

271 "concists of". I suggest checking the entire manuscript and systematically replacing "consists in" with "consists of".

372 This seems particularly risky for high towers in situations with strong convection. Maybe this could be stated explicitly rather than left vague as "further considerations".

374 "With respect to"

385 The post-dawn transition from a stable to a growing/convective boundary layer seems to be a moment that is particularly relevant here. Perhaps you could elaborate on this or at least prepare the user for its importance.

404 LOESS

414 loess

465 A key point: can you either justify this or evaluate its importance?

470 You are careful to describe the methods for the eddy covariance calculations. However, whereas the determination of the storage terms is not trivial, you provide no methods.

484 The meaning of the subscript (i=1,...,6) is not at all clear to me. What are the differences between the six cases?

485 "was run"

497 Figure 2 has no units on either axis. Also, whereas this figure has labels for every panel, many of the following figures do not (but should).

516 This is hardly a stand-alone figure (i.e., a reader would need to search broadly through the text in order to understand it). One might easily suppose that LSR means

"least-squares regression", and not a "low signal resolution" text. Again, the subscripts are not clear. What is the difference between i=1 and i=4 (both of which give phi values of 0.9). Do "A-R" and "Amp Res" refer to the same thing? Generally speaking, I think that the average reader would initially obtain no information from this figure, and would become bewildered after studying it for ten minutes.

519 "n0 indicates the percentage"

529 Does size 6000 mean 10 minutes of data?

531 Which 15% were replaced? I don't see a change in variance in Figure 5 (whose panels are not labelled, but I refer to Figure 5f

536 "was run"

549 the proposed tests were more selective

553 identify the same issue, which is why the ModEr statements

572 what are the units of the values 0.0004 and 0.004?

573 what are the units of the values 0.025 and 0.25?

573 change "Exemplary" to "Representative".

603 Figure 9 has no units. Also, understanding the meaning of FW96 and M98 would require the reader to have access to older papers that may not be readily available. When a measure is used to compare against a threshold, that variable should be explicitly defined (so as to make the figure stand alone).

634 The appearance of references (to the works of Cleveland, Lomb, and Scargle) in the Results section indicates poor structure. All of these tests should have been previously defined in the methods section.

639 The y-axis labels of Figure 13 are particularly weak. What is "c" to which NEE has been added prior to computing a logarithm? What is "D"? The term "Irregular" is nowhere precisely defined (although it appears to derive from the work of Lomb/Scargle). What are its units? Also, the choice of max/min for the y-axes of the LRT and SRT variables (likewise not clear from the figure) are poor and give no idea regarding the ranges of these variables.

643 "Previous research (refs) has highlighted"

648 Figure 14: Maybe change "DoY" to "Day of Year". To what year does it refer? Also, change "Density" to "Probability density".

---

## Referee Comment (RC2) · Anonymous Referee #2 · 7 Oct 2019

General comments This study presents a novel scheme for quality control (QC) of eddy-covariance data, which is a very relevant topic for the readership of this journal. Especially, since more and more data become freely available and are being used in large-scale synthesis studies, a sound and robust data cleaning procedure is needed. This is certainly not the first attempt to provide such a method, and a number of common existing methods are cited, but this new method is somewhat innovative, since it separates the quality tests from data rejection criteria more rigorously that other methods. This allows for more flexibility in the selection of test algorithms, so that future developments can be integrated more easily. While I find the data actual QC algorithm logical and coherent, I find it very bold to assume that a random uncertainty estimate served as the only quality indicator eddy-covariance data, as e.g. stated in the conclusion. More precisely, I have the following two concerns: 1.) I agree, that systematic errors should either be avoided or corrected for. However, the uncertainty of a flux estimate may increase as a result of a flux corrections (because the estimate is partially modelled and not measured, particularly as a result of spectral corrections). How can this be included? 2.) Moreover, the spatial representativeness of a flux estimate, cannot easily be accounted for in a random error estimate. What if there is a mixed land use within the flux footprint? This issue needs to be addressed in some way.

Detailed comments L98: . . . because it would be extremely time consuming L122: I disagree that the random uncertainty is sufficient to characterize eddy-covariance data, for the above-mentioned reasons. L140: I disagree with this statement. A bias can at least indirectly be determined using the energy balance closure. L461: This might be a better reference for the Selhausen site: (Schmidt et al., 2012) because it is only one of several sites that are used as an example in Mauder et al. (2013), and the correct Site-ID is DE-RuS

Schmidt, M., Reichenau, T. G., Fiener, P. and Schneider, K.: The carbon budget of a winter wheat field: An eddy covariance analysis of seasonal and inter-annual variability, Agric. For. Meteorol., 165, 114–126

---

## Author Comment (AC1) · 8 Nov 2019

We thank Andrew Kowalski (AK hereafter) for constructive comments and suggestions that helped us revise and clarify key concepts and the way we approached them in this study.

In the following, our replies to the referee comments.

[Figure]
**AK** - *The authors present a procedure for the "cleaning" of raw data from eddy covariance flux towers. If effective, such a procedure will be of great value. The procedure described is founded partially in previous studies of quality control for flux tower data, and partially based on statistical analyses that can be considered novel in this context. I am no expert in statistical analyses, but the approach put forth seems to be valid and defensible. The paper is well written, although I offer some minor suggestions intended for its improvement. My main problem with the paper is that the evaluation of the proposed scheme is based on synthetic time series, and I am left wondering whether such "data" are representative in terms of the types of problems that can be encountered in real, field data. If there were a means of assessing this key question, then I would suggest this as a "major" revision issue. In any event, however, I believe that the manuscript merits publication.*

**Authors' reply** - We understand and share the concerns raised regarding the use of synthetic data. However, the use of Monte Carlo simulations is attractive and constitutes a powerful and objective tool for the evaluation of methods.

Simulations allow, in fact, to get a full control of (1) the time series dynamics (since the simulated autoregressive processes are stationary with a pre-fixed, reference correlation), (2) the presence of a specific source of systematic error, and (3) the uncertainty due to random error component. As a consequence, a proper evaluation of the bias effect caused by systematic errors and a more objective performance evaluation of the tests involved for their detection, becomes feasible. Such evaluations are difficult to achieve with real, EC field data because the reference "true" value is unknown, therefore it is not possible to properly quantificate the bias effect, and replicates are not available, making it difficult the evaluation of the uncertainty associated with the estimates (either covariances and test statistics) due to the random error component.

We are aware that the model generating the simulated data differs from the actual data generating process. On the other hand, it is not an easy task to develop a stochastic

simulation model so complex as to include all the sources of error, either systematic or random, that are present in real-word EC time-series data. Building on the principle that "all models are wrong but some models are useful", we opted for a "useful" simulation model, having a simple dynamics that can reproduce, at least in part, the behavior of real EC data. If a QC test exhibits high false-positive and/or false-negative rates in such simple scenarios, it is most unlikely it will work properly with real data. In this case, the test would likely be methodologically robust but unusable in practice, affected by over-inflated type-I and type-II errors.

Thus, we are also aware that the robustness of our proposal needs to be investigated further. This paper is a starting point aiming at presenting a new strategy and a new framework that could hopefully stimulate innovative research on more complex EC data simulation designs, as well as on new and more robust QC tests for EC data to be used in the context of our data cleaning procedure.

In the revised version of the paper we will add a more clear note of caution on the fact that the evaluation of QC tests was performed on simulated time series affected only by a limited number of sources of systematic and random error.

**AK** - Specific comments (by line number)

- 38 "An integral part of the EC?"

- 83 "classified into partially versus completely data-driven"

- 87 "Such a call"

- 93 sacrifice

- 98 "because it is extremely"

- 100 "false-positive rates"

- 105 "selected, the results of these tests"

- 135 "proposed set, and iv) results in"

- 142 "a set of tests"

- 200 "with some instruments, especially older models or often"

- 374 "With respect to"

- 404 LOESS 414 loess

- 485 "was run"

- 519 "n0 indicates the percentage"

- 536 "was run"

- 549 the proposed tests were more selective

- 553 identify the same issue, which is why the ModEr statements

- 573 change "Exemplary" to "Representative".

- 643 "Previous research (refs) has highlighted"

- 648 Figure 14: Maybe change "DoY" to "Day of Year". To what year does it refer? Also, change "Density" to "Probability density"

**Authors' reply** - We thank AK for the above suggestions and corrections that will be implemented in the new version of the paper.

**AK** - 82 *This statement strikes is as excessively bold and over-simplified. For example, a random error that were to consistently reduce the covariance at a frequency of 0.01Hz (by introducing random noise at that frequency) would cause flux underestimation for the corresponding eddies. It could well be that these frequencies matter greatly for (convective) eddy transport during daytime, and far less so at night. The result would be an overestimation of NEE over the long term.*

**Authors' reply** - We agree with the reviewer statement that the presence of a specific source of error as the one reported in the example, could attenuate covariances and, consequently, introduce long-term biases.

We shall clarify that the distinction between random and systematic error affecting half-hourly flux estimates is based on the effects of the source of error on the quantity of interest and is thus not strictly linked to its features. In other words, the question to answer is: is the source of error responsible to introduce bias or to increase the uncertainty of the quantity of interest (e.g. covariance)?

With this definition, if the presence of some source of error is responsible for attenuating flux covariance estimates, then the source of error is systematic, even if the error component is a noise term having similar characteristic to random data. If, instead, the presence of some source of error is responsible for increasing the uncertainty associated with flux covariance estimates (i.e. standard deviation) then the source of error is classified as random. We are aware that, in practice, it is difficult to distinguish between random and systematic errors because some source of error can have both a random and a systematic component, there are no reference values to evaluate the presence of bias, and there are not replicates to consistently quantificate the random uncertainty.

However, we consider the above-mentioned classification more suitable to characterize the multitude of errors affecting eddy-covariance time series. We will better clarify these concepts in the introductory section of the revised version of the manuscript.

**AK** - 200 *Table 1: if the terms "fraction of missing records (FMR)" and "longest gap duration (LGD)" are defined with the acronyms in parentheses as in this sentence, then it makes it easier for the reader to find them when they encounter them for the first time in the text.*

**Authors' reply**. Thanks for the suggestion, we agree and so we will introduce the definition and acronym in the text.

**AK** - 248 *Again, I find greatly exaggerated the claim the random errors due not affect covariances. This depends on the magnitudes of the signal/noise variances, and also on their durations during the averaging period.*

**Authors' reply** - As explained before, the "unavoidable random measurement error" we are referring is the one that, by definition, is not responsible for introducing bias in flux estimates. To avoid confusion, we call them simply random error and add a reference to the definition of random/systematic error described in the introductory section.

**AK** - 372 *This seems particularly risky for high towers in situations with strong convection. Maybe this could be stated explicitly rather than left vague as "further considerations".*

**Authors' reply** - We rephrase the sentence as suggested, however what is reported is the suggestion by Mahrt in his paper proposing the test.

**AK** - 385 *The post-dawn transition from a stable to a growing/convective boundary layer seems to be a moment that is particularly relevant here. Perhaps you could elaborate on this or at least prepare the user for its importance.*

**Authors' reply** - It is true, although the change of conditions happening in the post-dawn period doesn't lead automatically to the presence of outliers (as per our definition). Nonetheless, we will add this as an example.

**AK** - 465 *A key point: can you either justify this or evaluate its importance?*

**Authors' reply** - We understand the importance as also discussed in the first part of the replies. We will add a justification that is based on the fact that a proper validation can only be done by comparing the results with true reference values. This is not possible with real, EC field data because we don't know with certainty whether data are error-free or not. For this reason the performance evaluation is carried out via Monte Carlo studies where we have a full control of time series dynamics, error contribution and its bias effects on the quantity of interest. The use of real data gives an idea of the impact the proposed schemes has in cleaning EC data and can help to understand which is the main source of systematic error affecting EC flux variables.

**AK** - 470 *You are careful to describe the methods for the eddy covariance calculations. However, whereas the determination of the storage terms is not trivial, you provide no methods.*

**Authors' reply** - We will add more information about the storage calculation (the best available since not all the selected sites had profile measurements available). However it is important to note that the effect of the storage doesn't influence the QC procedure

that is based on the turbulent data time series. The storage term does have an effect on the computed NEE that is used to identify outliers, however the effect is minimal. We will add however an explicit sentence on the fact that the storage term must be correctly measured and calculated, including proper references.

**AK** - 484 *The meaning of the subscript (i=1,...,6) is not at all clear to me. What are the differences between the six cases?*

**Authors' reply** - We recognize that the six cases are not clearly defined (the definition is in different parts of the text). We will add a detailed description of the 6 scenarios in the new version of the paper.

**AK** - 497 *Figure 2 has no units on either axis. Also, whereas this figure has labels for every panel, many of the following figures do not (but should).*

**Authors' reply** - We will add labels on all the axes of all the figures. About the letters, we used them only when needed to refer to specific subplots in the text, however it is not a problem to add them in all the figures.

**AK** - 516 *This is hardly a stand-alone figure (i.e., a reader would need to search broadly through the text in order to understand it). One might easily suppose that LSR means "least-squares regression", and not a "low signal resolution" test. Again, the subscripts are not clear. What is the difference between i=1 and i=4 (both of which give phi values of 0.9). Do "A-R" and "Amp Res" refer to the same thing? Generally speaking, I think that the average reader would initially obtain no information from this figure, and would become bewildered after studying it for ten minutes.*

**Authors' reply** - We agree with the reviewer on the complexity of the figure. However, the plots illustrated herein are crucial as they summarize important information. In order to improve readability and clearness of the figure, we will use labels, uniform the acronyms in the figure and throughout the text and improve the caption. The figure will then be self explaining and easy to get in the main message.

**AK** - 529 *Does size 6000 mean 10 minutes of data?*

**Authors' reply** - Yes, in case of time series sampled at 10Hz scanned frequency. We will clarify this in the text.

**AK** - 531 *Which 15% were replaced? I don't see a change in variance in Figure 5 (whose panels are not labelled, but I refer to Figure 5f.*

**Authors' reply** - It is 15% of simulated data points randomly selected, and the change in variance is evident when compared to the uncontaminated time series, which is however not shown. It will be added as a subplot of Figure 5 in the revised version of the manuscript. We will also add labels to each panel and provide more details in the figure caption.

**AK** - 572 *What are the units of the values 0.0004 and 0.004?*

**Authors' reply** - These values, as slope coefficients of deterministic time trend contaminating the simulated stationary autoregressive processes are dimensionless. They quantify the change in the mean level, per unit of time.

In case of time series of length $n = 18000$, as in EC raw data sampled at 10Hz in an

average period of 30 minutes, a slope coefficient equal to 0.0004 (0.004) increases the mean level of 7.2 (72) units of the response variable, respectively (e.g. 7.2 ms$^{-1}$ for vertical wind speed, 72 $\mu$mol/mol for $CO_2$ concentrations, 72 K for sonic temperature).

We will better explain their meaning in the revised version of the manuscript.

**AK** - 573 *What are the units of the values 0.025 and 0.25?*

**Authors' reply** - The units of the pre-fixed standard deviation values used to generate stochastic trends contaminating the simulated stationary autoregressive processes are dimensionless.

Stochastic trends were generated as cumulative sum of $n$ independent random variables normally distributed with mean 0 and standard deviation (s) equal to 0.025 or 0.25. Consequently, the sum of such variables is Normal with mean 0 and standard deviation $\sigma = \sqrt{n} \cdot s$.

In case of time series of length $n = 18000$, as in EC raw data sampled at 10Hz in an average period of 30 minutes, $s$ equal to 0.025 (0.25) generates an ensemble of stochastic trajectories, responsible for non-stationary conditions, whose standard deviation at $n = 18000$ is $\sigma = 3.35(33.5)$ units of the response variable, respectively (e.g. ms$^{-1}$ for vertical wind speed, $\mu$mol/mol for $CO_2$ concentrations, K for sonic temperature).

We will better explain their meaning in the revised version of the manuscript.

**AK** - 603 *Figure 9 has no units. Also, understanding the meaning of FW96 and M98 would require the reader to have access to older papers that may not be readily available. When a measure is used to compare against a threshold, that variable should be explicitly defined (so as to make the figure stand alone).*

**Authors' reply** - We agree that figures should be stand alone and so we will review all labels and captions in order to give all the elements to the reader to understand the concept and meaning. Specifically on Figure 9, we will report in the caption that test statistics (x-axis) and correlation (y-axis) between simulated time series are dimensionless. As the meaning of the stationary tests and the choice of threshold values were introduced in Section 2.2.3, we will provide more details and report the reference to the Section 2.2.3 in the figure caption.

**AK** - 634 *The appearance of references (to the works of Cleveland, Lomb, and Scargle) in the Results section indicates poor structure. All of these tests should have been previously defined in the methods section.*

**Authors' reply** - The references mentioned by the reviewer are present in the introduction and material and methods sections, and were reported also in the Result section aiming at improving its clearness. However, we agree that this may reduce the readability of this section. We will remove them from the Results section in the revised version of the paper.

**AK** - 639 *The y-axis labels of Figure 13 are particularly weak. What is "c" to which NEE has been added prior to computing a logarithm? What is "D"? The term "Irregular" is nowhere precisely defined (although it appears to derive from the work of Lomb/Scargle). What are its units? Also, the choice of max/min for the y-axes of the LRT and SRT variables (likewise not clear from the figure) are poor and give no idea regarding the ranges of these variables.*

**Authors' reply** - Recognizing the length and complexity of the paper we removed some

text and explanations, which ultimately made the Figure unclear for the readers. We will review labels and caption of Figure 13. In particular:

- c is a constant added to flux variable to ensure the argument of log is positive;

- D, LRT, SRT and Irregular indicate the components of the STL decomposition, as described in detail in Section 2.3.1 (we will report the details also in the figure caption);

- units: log-transformed variables are dimensionless. In Figure 13, only original NEE time series and the Irregular component (after inverse log-transformation) are in $\mu$mol CO$_2$ m$^{-2}$ s$^{-1}$.

- y-axis range: we will adjust it.

**AK** - 648 *Figure 14: Maybe change "DoY" to "Day of Year". To what year does it refer? Also, change "Density" to "Probability density".*

**Authors' reply** - Thank you for this comment. We will change the labels accordingly.

---

## Author Response (AR1)

**Responses to the Associate Editor**

**Associate Editor** -*In this manuscript, Vitale et al present a new cleaning method for eddy covariance data. Both reviewers were positive about the work and I have decided that major revisions are required. R1 wonders about the transferability of the method beyond purely synthetic data, I feel this is a fair question and wonder if the authors cannot at least partially address this with some further text if not new analysis? R2 asks two important questions that should be addressed, preferably, in a revised discussion. Can I also suggest:*

1. *the abstract lacks a clear statement about the problems this method will solve, it is in my eyes, a little too vague. A firm example would be preferable.*

2. *I did not see a statement about code sharing. Have I missed it, this would seem very important if the authors are proposing a new method to the community.*

**Authors' reply** - Thank you for these comments. In the revised version of the manuscript, all the Reviewers' requests have been implemented and addressed. A detailed description of changes is given in the following.

We have also implemented and addressed your requests. In particular, the Abstract has been modified as you suggested and the proposed data cleaning has been implemented in RFlux, an R software package which is publicly available from a GitHub repository. Information about RFlux are given on page 7, lines 187-192 and on page 35, line 735 (Code and data availability)

**Responses to Andrew Kowalski**

We thank Andrew Kowalski (AK hereafter) for constructive comments and suggestions that helped us revise and clarify key concepts and the way we approached them in this study.

In the following, our replies to the referee comments.

**AK** - *The authors present a procedure for the "cleaning" of raw data from eddy covariance flux towers. If effective, such a procedure will be of great value. The procedure described is founded partially in previous studies of quality control for flux tower data, and partially based on statistical analyses that can be considered novel in this context. I am no expert in statistical analyses, but the approach put forth seems to be valid and defensible. The paper is well written, although I offer some minor suggestions intended for its improvement. My main problem with the paper is that the evaluation of the proposed scheme is based on synthetic time series, and I am left wondering whether such "data" are representative in terms of the types of problems that can be encountered in real, field data. If there were a means of assessing this key question, then I would suggest this as a "major" revision issue. In any event, however, I believe that the manuscript merits publication.*

**Authors' reply** - We understand and share the concerns raised regarding the use of synthetic data. However, the use of Monte Carlo simulations is attractive and constitutes a powerful and objective tool for the evaluation of methods.

Simulations allow, in fact, to get a full control of (1) the time series dynamics (since the simulated autoregressive processes are stationary with a pre-fixed, reference correlation), (2) the presence of a specific source of systematic error, and (3) the uncertainty due to random error component. As a consequence, a proper evaluation of the bias effect caused by systematic errors and a more objective performance evaluation of the tests involved for their detection, becomes feasible. Such evaluations are difficult to achieve with real, EC field data because the reference "true" value is unknown, therefore it is not possible to properly quantificate the bias effect, and replicates are not available, making it difficult the evaluation of the uncertainty associated with the estimates (either covariances and test statistics) due to the random error component.

We are aware that the model generating the simulated data differs from the actual data generating process. On the other hand, it is not an easy task to develop a stochastic simulation model so complex as to include all the sources of error, either systematic or random, that are present in real-word EC time-series data. Building on the principle that "all models are wrong but some models are useful", we opted for a "useful" simulation model, having a simple dynamics that can reproduce, at least in part, the behavior of real EC data. If a QC test exhibits high false-positive and/or false-negative rates in such simple scenarios, it is most unlikely it will work properly with real data. In this case, the test would likely be methodologically robust but unusable in practice, affected by over-inflated type-I and type-II errors.

Thus, we are also aware that the robustness of our proposal needs to be investigated further. This paper is a starting point aiming at presenting a new strategy and a new framework that could hopefully stimulate innovative research on more complex EC data simulation designs, as well as on new and more robust QC tests for EC data to be used in the context of our data cleaning procedure.

In the revised version of the paper on page 34, lines 718-730, we have emphasized the fact that the evaluation of QC tests was performed on simulated time series affected only by a limited number of sources of systematic and random error.

**AK** - Specific comments (by line number)

- 38 "An integral part of the EC?"

- 83 "classified into partially versus completely data-driven"

- 87 "Such a call"

- 93 sacrifice

- 98 "because it is extremely"

- 100 "false-positive rates"

- 105 "selected, the results of these tests"

- 135 "proposed set, and iv) results in"

- 142 "a set of tests"

- 200 "with some instruments, especially older models or often"

- 271 "consist of "

- 374 "With respect to"

- 404 LOESS 414 loess

- 485 "was run"

- 519 "n0 indicates the percentage"

- 536 "was run"

- 549 the proposed tests were more selective

- 553 identify the same issue, which is why the ModEr statements

- 573 change "Exemplary" to "Representative".

- 643 "Previous research (refs) has highlighted"

- 648 Figure 14: Maybe change "DoY" to "Day of Year". To what year does it refer? Also, change "Density" to "Probability density"

**Authors' reply** - We thank AK for the above suggestions and corrections that have been implemented in the revised version of the paper.

**AK** - 82 *This statement strikes is as excessively bold and over-simplified. For example, a random error that were to consistently reduce the covariance at a frequency of 0.01Hz (by introducing random noise at that frequency) would cause flux underestimation for the corresponding eddies. It could well be that these frequencies matter greatly for (convective) eddy transport during daytime, and far less so at night. The result would be an overestimation of NEE over the long term.*

**Authors' reply** - We agree with the reviewer statement that the presence of a specific source of error as the one reported in the example, could attenuate covariances and, consequently, introduce long-term biases.

We clarified that the distinction between random and systematic error affecting half-hourly flux estimates is based on the effects of the source of error on the quantity of interest and is thus not strictly linked to its features. In other words, the question to answer is: is the source of error responsible to introduce bias or to increase the uncertainty of the quantity of interest (e.g. covariance)?

With this definition, if the presence of some source of error is responsible for attenuating flux covariance estimates, then the source of error is systematic, even if the error component is a noise term having similar characteristics to random data. If, instead, the presence of some source of error is responsible for increasing the uncertainty associated with flux covariance estimates (i.e. standard deviation) then the source of error is classified as random. We are aware that, in practice, it is difficult to distinguish between random and systematic errors because some source of error can have both a random and a systematic component, there are no reference values to evaluate the presence of bias, and there are not replicates to consistently quantificate the random uncertainty.

However, we consider the above-mentioned classification more suitable to characterize the multitude of errors affecting eddy-covariance time series. We have better clarified these concepts in the introductory section (page 3, lines 68-84) of the revised version of the manuscript.

**AK** - 200 *Table 1: if the terms "fraction of missing records (FMR)" and "longest gap duration (LGD)" are defined with the acronyms in parentheses as in this sentence, then it makes it easier for the reader to find them when they encounter them for the first time in the text.*

**Authors' reply**. Thanks for this comment. We have introduced definitions and acronyms on page 9, line 238, and in Table 1 (page 8) as you suggested.

**AK** - 248 *Again, I find greatly exaggerated the claim the random errors due not affect covariances. This depends on the magnitudes of the signal/noise variances, and also on their durations during the averaging period.*

**Authors' reply** - As explained before, the "unavoidable random measurement error" we are referring is the one that, by definition, is not responsible for introducing bias in flux estimates. To avoid confusion, we call them simply random error and have added a reference to the definition of random/systematic error described in the introductory section (page 10, lines 251-252).

**AK** - 372 *This seems particularly risky for high towers in situations with strong convection. Maybe this could be stated explicitly rather than left vague as "further considerations".*

**Authors' reply** - We have rephrased the sentence as suggested (page 14, lines 376-377), however what is reported is the suggestion by Mahrt in his paper proposing the test.

**AK** - 385 *The post-dawn transition from a stable to a growing/convective boundary layer seems to be a moment that is particularly relevant here. Perhaps you could elaborate on this or at least prepare the user for its importance.*

**Authors' reply** - It is true, although the change of conditions happening in the post-dawn period doesn't lead automatically to the presence of outliers (as per our definition). Nonetheless, we have added this as an example on page 14, lines 387-388.

**AK** - 465 *A key point: can you either justify this or evaluate its importance?*

**Authors' reply** - We understand the importance as also discussed in the first part of the replies. In the revised version, we have provided a justification which is based on the fact that a proper validation can only be done by comparing the results with true reference values. This is not possible with real, EC field data because we don't know with certainty whether data are error-free or not. For this reason the performance evaluation is carried out via Monte Carlo studies where we have a full control of time series dynamics, error contribution and its bias effects on the quantity of interest. The use of real data gives an idea of the impact the proposed schemes has in cleaning EC data and can help to understand which is the main source of systematic error affecting EC flux variables. See the revised version of the manuscript on page 17, lines 472-476.

**AK** - 470 *You are careful to describe the methods for the eddy covariance calculations. However, whereas the determination of the storage terms is not trivial, you provide no methods.*

**Authors' reply** - More details about the storage calculation are now provided on page 17, lines 483-486, of the revised version of the manuscript.

**AK** - 484 *The meaning of the subscript (i=1,...,6) is not at all clear to me. What are the differences between the six cases?*

**Authors' reply** - We recognize that the six cases are not clearly defined (the definition is in different parts of the text). We have added a detailed description of the 6 scenarios in the new version of the paper on page 18, lines 500-505.

**AK** - 497 *Figure 2 has no units on either axis. Also, whereas this figure has labels for every panel, many of the following figures do not (but should).*

**Authors' reply** - We have add labels on all the axes of all the figures. About the letters, we used them only when needed to refer to specific subplots in the text.

**AK** - 516 *This is hardly a stand-alone figure (i.e., a reader would need to search broadly through the text in order to understand it). One might easily suppose that LSR means "least-squares regression", and not a "low signal resolution" test. Again, the subscripts are not clear. What is the difference between i=1 and i=4 (both of which give phi values of 0.9). Do "A-R" and "Amp Res" refer to the same thing? Generally speaking, I think that the average reader would initially obtain no information from this figure, and would become bewildered after studying it for ten minutes.*

**Authors' reply** - We agree with the reviewer on the complexity of Figure 3. However, the plots illustrated herein are crucial as they summarize important information. In order to improve readability and clearness of the figure, we have used labels, uniformed the acronyms in the figure (Fig. 3, page 19) and throughout the text and improved the corresponding caption.

**AK** - 529 *Does size 6000 mean 10 minutes of data?*

**Authors' reply** - Yes, in case of time series sampled at 10 Hz scanned frequency. We have better clarified this in the text on page 20, line 551.

**AK** - 531 *Which 15% were replaced? I don't see a change in variance in Figure 5 (whose panels are not labelled, but I refer to Figure 5f.*

**Authors' reply** - It is 15% of simulated data points randomly selected, and the change in variance is evident when compared to the uncorrupted time series which is now shown in the revised version of Figure 5 (page 22). Note also that the AR process has been simulated to mimic a sonic temperature time series. We have also added labels to each panel and provided more details in the corresponding figure caption.

**AK** - 572 *What are the units of the values 0.0004 and 0.004?*

**Authors' reply** - These values, as slope coefficients of deterministic time trend contaminating the simulated stationary autoregressive processes, are dimensionless. They quantify the change in the mean level, per unit of time.

In case of time series of length $n = 18000$, as in EC raw data sampled at 10Hz in an average period of 30 minutes, a slope coefficient equal to 0.0004 (0.004) increases the mean level of 7.2 (72) units of the response variable, respectively (e.g. 7.2 ms$^{-1}$ for vertical wind speed, 72 $\mu$mol mol$^{-1}$ for $CO_2$ concentrations, 72 K for sonic temperature).

We have better explained their meaning in the revised version of the manuscript (page 25, lines 598-602).

**AK** - 573 *What are the units of the values 0.025 and 0.25?*

**Authors' reply** - The units of the pre-fixed standard deviation values used to generate stochastic trends contaminating the simulated stationary autoregressive processes are dimensionless.

Stochastic trends were generated as cumulative sum of $n$ independent random variables normally distributed with mean 0 and standard deviation (s) equal to 0.025 or 0.25. Consequently, the sum of such variables is Normal with mean 0 and standard deviation $\sigma = \sqrt{n} \cdot s$.

In case of time series of length $n = 18000$, as in EC raw data sampled at 10Hz in an average period

of 30 minutes, $s$ equal to 0.025 (0.25) generates an ensemble of stochastic trajectories, responsible for non-stationary conditions, whose standard deviation at $n = 18000$ is $\sigma = 3.35(33.5)$ units of the response variable, respectively (e.g. ms$^{-1}$ for vertical wind speed, $\mu$mol mol$^{-1}$ for CO$_2$ concentrations, K for sonic temperature).

We have better explained their meaning in the revised version of the manuscript (page 25, lines 603-608).

**AK** - 603 *Figure 9 has no units. Also, understanding the meaning of FW96 and M98 would require the reader to have access to older papers that may not be readily available. When a measure is used to compare against a threshold, that variable should be explicitly defined (so as to make the figure stand alone).*

**Authors' reply** - We agree that figures should be stand alone and so we have reviewed all labels and captions in order to give all the elements to the reader to understand the concept and meaning. Specifically on Figure 9 (page 27), we have added units and provided more details in the corresponding figure caption.

**AK** - 634 *The appearance of references (to the works of Cleveland, Lomb, and Scargle) in the Results section indicates poor structure. All of these tests should have been previously defined in the methods section.*

**Authors' reply** - The references mentioned by the reviewer are present in the introduction and material and methods sections, and were reported also in the Result section aiming at improving its clearness. However, we agree that this may reduce the readability of this section. We have then removed them from the Results section in the revised version of the paper.

**AK** - 639 *The y-axis labels of Figure 13 are particularly weak. What is "c" to which NEE has been added prior to computing a logarithm? What is "D"? The term "Irregular" is nowhere precisely defined (although it appears to derive from the work of Lomb/Scargle). What are its units? Also, the choice of max/min for the y-axes of the LRT and SRT variables (likewise not clear from the figure) are poor and give no idea regarding the ranges of these variables.*

**Authors' reply** - Recognizing the length and complexity of the paper we removed some text and explanations, which ultimately made the Figure unclear for the readers. We have reviewed labels and caption of Figure 13 (page 32). In particular:

- c is a constant added to flux variable to ensure the argument of log is positive;

- D, LRT, SRT and Irregular indicate the components of the STL decomposition, as described in detail in Section 2.3.1 (we have reported the details also in the figure caption);

- units: log-transformed variables are dimensionless. In Figure 13, only original NEE time series and the Irregular component (after inverse log-transformation) are in $\mu$mol CO$_2$ m$^{-2}$ s$^{-1}$.

- y-axis range is now adjusted.

**Responses to Reviewer#2**

**Referee general comment** - *This study presents a novel scheme for quality control (QC) of eddy-covariance data, which is a very relevant topic for the readership of this journal. Especially, since more and more data become freely available and are being used in large-scale synthesis studies, a sound and robust data cleaning procedure is needed. This is certainly not the first attempt to provide such a method, and a number of common existing methods are cited, but this new method is somewhat innovative, since it separates the quality tests from data rejection criteria more rigorously than other methods. This allows for more flexibility in the selection of test algorithms, so that future developments can be integrated more easily.*

**Authors' reply** - Thank you for this kind reply and consideration. We appreciate that she/he caught the essence of the proposed approach. In the following, our replies to the referee comments.

**Referee comment** - *While I find the data actual QC algorithm logical and coherent, I find it very bold to assume that a random uncertainty estimate served as the only quality indicator eddy-covariance data, as e.g. stated in the conclusion.*

**Authors' reply** -We didn't state that this is the only quality indicator indeed. Rather, we state that, given an unbiased flux covariance estimate (as reported on page 34, lines 703:704), a consistent estimate of the random uncertainty would provide an objective criterion for evaluating the quality of data. We have better clarified this concept in the Conclusion section of the revised version of the manuscript.

**Referee comment** - *More precisely, I have the following two concerns*:

1. *I agree, that systematic errors should either be avoided or corrected for. However, the uncertainty of a flux estimate may increase as a result of a flux corrections (because the estimate is partially modelled and not measured, particularly as a result of spectral corrections). How can this be included?*

   **Authors' reply** - Although an interesting topic, this work is not focused on the evaluation of flux correction methods neither on the evaluation of the error propagation. This paper aims at describing a flexible and robust data cleaning procedure for eddy covariance flux measurements. Of course, the uncertainty associated to flux estimates should take into account the contribution of all possible sources, including those related to flux correction procedures. However, the availability of "high-quality" datasets constitutes an essential prerequisite to increase the robustness and reduce the uncertainty of the results of flux correction procedures, for example those used for the estimation of spectral correction factors. In this perspective, the application of robust data cleaning procedures, as the one proposed in this work, can help to achieve consistent estimates of correction parameters and, consequently, less biased and uncertain flux estimates. We have added such considerations in the revised version of the manuscript (see lines 713-717 on page 34).

2. *Moreover, the spatial representativeness of a flux estimate, cannot easily be accounted for in a random error estimate. What if there is a mixed land use within the flux footprint? This issue needs to be addressed in some way.*

**Authors' reply** -We agree with the reviewer's concern, but this is mainly an issue related to quality assurance (QA) step, i.e. a problem of the site's characteristics with respect to the requirements of the eddy covariance method. This is out of the scope of the present manuscript. However, the contribution of spatial sampling in random uncertainty estimates has been mentioned in the revised version (see lines 710-713 on page 34).

**Referee comment** - L98: *. . . because it would be extremely time consuming.*

**Authors' reply** -Thank you for this suggestion. It has been taken into consideration in the revised version of the manuscript.

**Referee comment** - L122: *I disagree that the random uncertainty is sufficient to characterize eddy-covariance data, for the above-mentioned reasons.*

**Authors' reply** -We tried to clarify above the point and where the concept applies in this case. The assumption underlying this statement relies to the availability of unbiased flux estimates and, also, the availability of consistent estimates of random uncertainty. We have better clarified these concepts both in the introductory (page 4, lines 68:84) and in the concluding section (page 34, lines 705:717) of the revised version of the manuscript. See also answer to reviewer 1 reported below.

***Andrew Kowalski comment***: *L82 This statement strikes is as excessively bold and over-simplified. For example, a random error that were to consistently reduce the covariance at a frequency of 0.01Hz (by introducing random noise at that frequency) would cause flux underestimation for the corresponding eddies. It could well be that these frequencies matter greatly for (convective) eddy transport during daytime, and far less so at night. The result would be an overestimation of NEE over the long term.*

**Authors' reply to Andrew Kowalski comment.** We agree with the reviewer statement that the presence of a specific source of error as the one reported in the example, could attenuate covariances and, consequently, introduce long-term biases. We have clarified that the distinction between random and systematic error affecting half-hourly flux estimates is based on the effects of the source of error on the quantity of interest and is thus not strictly linked to its features. In other words, the question to answer is: is the source of error responsible to introduce bias or to increase the uncertainty of the quantity of interest (e.g. covariance)? With this definition, if the presence of some source of error is responsible for attenuating flux covariance estimates, then the source of error is systematic, even if the error component is a noise term having similar characteristic to random data. If, instead, the presence of some source of error is responsible for increasing the uncertainty associated with flux covariance estimates (i.e. standard deviation) then the source of error is classified as random.

We are aware that, in practice, it is difficult to distinguish between random and systematic errors because some source of error can have both a random and a systematic component, there are no reference values to evaluate the presence of bias, and there are not replicates to consistently quantificate the random uncertainty. However, we consider the above-mentioned classification more suitable to characterize the multitude of errors affecting eddy-covariance time series.

**Referee comment.** L140: *I disagree with this statement. A bias can at least indirectly be determined using the energy balance closure.*

**Authors' reply** - We respectfully disagree with the reviewer for this point. Although important, the energy balance closure is still an open issue in the eddy covariance methodology, as the many attempts to explain and justify it didn't fully succeed so far. That's why we believe it can only be used to indirectly suggest the presence of some unknown source of systematic error affecting flux estimates, but not to determine it, at least at half-hourly scale (and in fact there is no consensus about the fluxes correction procedure on the basis of the energy balance closure). In the revised version of the manuscript we have added some consideration (page 34, lines 716-717) about the use of additional analyses as indicative of the presence of undetected/unknown sources of systematic error.

**Referee comment.** L461: *This might be a better reference for the Selhausen site: (Schmidt et al., 2012) because it is only one of several sites that are used as an example in Mauder et al. (2013), and the correct Site-ID is DE-RuS Schmidt, M., Reichenau, T. G., Fiener, P. and Schneider, K.: The carbon budget of a winter wheat field: An eddy covariance analysis of seasonal and inter-annual variability, Agric. For. Meteorol., 165, 114–126*

**Authors' reply** - Thanks for the suggestion, we have reviewed the citations of all the sites and data and included specific references and acknowledgments.

[revised manuscript text omitted]

---

## Referee Report (RR1)

The authors have appropriately revised the originally submitted manuscript, and I believe the paper is nearly ready for publication. Nonetheless, I offer the following suggested revisions intended to improve the quality of the language used (by line number):

9. Change "lead" to either "leads" or "led", depending upon the preferred tense.

27. Change "explain/qualify" to "explaining/qualifying".

35. Change "associated to" to "of".

44. Delete "to edge".

65. Change "making sure sigma is as small as possible" to "minimizing sigma".

68. Change "source" to "sources".

127. Change "in depth" to "in-depth".

143. Change "at half-hourly scale" to "at a half-hourly scale".

280. Change "significantly" to "significant".

303. Change "measurement" to "measurements".

351. Change "expose to the risks of" to "risk".

376. Change "recommendation is to reject" to "recommended rejecting"

387. Change "cause" to "causes".

411. Maybe capitalize "loess"?

423. Change "was imposed" to "were imposed".

427. Change "is" to "has been" (?).

446. Change "simulate" to "simulated".

463. Change "error, details" to "error, the details".

465. Change "sites part" to "sites that are part".

473. Change "contribution" to "contributions".

474. Change "crucially" to "crucial".

484. Change "profiles" to "profile".

499. Change "to create the typical ramp structure" to "the creation of a typical ramp structure as is".

522. Change "showed a good" to "showed good".

529. Change "for sensible heat flux" to "for the sensible heat flux".

535. Change "evidences" to "evidence".

556. Change "to the malfunctioning" to "to malfunction".

565. Change "a low" to "poor".

622. Change "independently from the quantity at the numerator" to "independent of the numerator".

625. Change "the statistic exceeded" to "did the statistic exceed".

627. Change "at detecting" to "to detect".

630. Change "by Foken" to "of Foken".

650. Change "Section" to "section".

660. Delete "peak".

682. Change "was detected as outlier" to "were detected as outliers".

691. Delete "s about the presence".

692. Delete "s about the presence of sources".

693. Change "evidences" to "evidence".

696. Delete "the presence".

697. Delete "source of" and "is".

715. Change "associated to" to "of".

719. Delete "to get a".

721. Change "to random" to "to the random".

724. Change "the evaluation of" to "to evaluate".

736. Change "either systematic or" to "both systematic and".

730. What are type-I and type-II errors?

---

## Author Response (AR2)

**Responses to the Associate Editor**

Associate Editor - Dear Authors, two reviewers have now looked through your revised manuscript and I'm happy to recommend acceptance subject to some very minor final edits, congratulations. Reviewer 2 has kindly suggested a few minor improvements, which I am happy to check through once you have revised your manuscript.

**Authors' reply** - Dear Editor, we wish to thank you, Andrew Kowalski and the anonymous reviewer for the constructive comments and suggestions that have greatly improved the readability of the manuscript. In the revised version, all the Reviewers' requests have been implemented and addressed. A detailed description of changes is given in the following.

**Responses to Andrew Kowalski**

**AK** -The authors have appropriately revised the originally submitted manuscript, and I believe the paper is nearly ready for publication. Nonetheless, I offer the following suggested revisions intended to improve the quality of the language used (by line number):

- 9. Change "lead" to either "leads" or "led", depending upon the preferred tense.
- 27. Change "explain/qualify" to "explaining/qualifying".
- 35. Change "associated to" to "of".
- 44. Delete "to edge".
- 65. Change "making sure sigma is as small as possible" to "minimizing sigma".
- 68. Change "source" to "sources".
- 127. Change "in depth" to "in-depth".
- 143. Change "at half-hourly scale" to "at a half-hourly scale".
- 280. Change "significantly" to "significant".
- 303. Change "measurement" to "measurements".
- 351. Change "expose to the risks of" to "risk".
- 376. Change "recommendation is to reject" to "recommended rejecting"
- 387. Change "cause" to "causes".
- 411. Maybe capitalize "loess"?
- 423. Change "was imposed" to "were imposed".
- 427. Change "is" to "has been" (?).
- 446. Change "simulate" to "simulated".
- 463. Change "error, details" to "error, the details".
- 465. Change "sites part" to "sites that are part".
- 473. Change "contribution" to "contributions".
- 474. Change "crucially" to "crucial".
- 484. Change "profiles" to "profile".
- 499. Change "to create the typical ramp structure" to "the creation of a typical ramp structure as is".

- 522. Change "showed a good" to "showed good".
- 529. Change "for sensible heat flux" to "for the sensible heat flux".
- 535. Change "evidences" to "evidence".
- 556. Change "to the malfunctioning" to "to malfunction".
- 565. Change "a low" to "poor".
- 622. Change "independently from the quantity at the numerator" to "independent of the numerator".
- 625. Change "the statistic exceeded" to "did the statistic exceed".
- 627. Change "at detecting" to "to detect".
- 630. Change "by Foken" to "of Foken".
- 650. Change "Section" to "section".
- 660. Delete "peak".
- 682. Change "was detected as outlier" to "were detected as outliers".
- 691. Delete "s about the presence".
- 692. Delete "s about the presence of sources".
- 693. Change "evidences" to "evidence".
- 696. Delete "the presence".
- 697. Delete "source of" and "is".
- 715. Change "associated to" to "of".
- 719. Delete "to get a".
- 721. Change "to random" to "to the random".
- 724. Change "the evaluation of" to "to evaluate".
- 726. Change "either systematic or" to "both systematic and".

Authors' reply - We thank Andrew Kowalski. The above suggestions and corrections have been implemented in the revised version of the paper.

**AK** - L730. What are type-I and type-II errors?

**Authors' reply** - A type-I error is equivalent to a false positive, and a type-II error is equivalent to a false negative. To avoid confusion and in accordance with the terminology used in the rest of the manuscript, we changed them to false positive and false negative errors.

[revised manuscript text omitted]